# Efficient Adaptive Experimentation with Noncompliance

**Miruna Oprescu**
Cornell University
amo78@cornell.edu

**Brian M Cho**
Cornell University
bmc233@cornell.edu

**Nathan Kallus**
Cornell University & Netflix
kallus@cornell.edu

## Abstract

We study the problem of estimating the average treatment effect (ATE) in adaptive experiments where treatment can only be encouraged—rather than directly assigned—via a binary instrumental variable. Building on semiparametric efficiency theory, we derive the efficiency bound for ATE estimation under arbitrary, history-dependent instrument-assignment policies, and show it is minimized by a variance-aware allocation rule that balances outcome noise and compliance variability. Leveraging this insight, we introduce AMRIV—an **A**daptive, **M**ultiply-**R**obust estimator for **I**nstrumental-**V**ariable settings with variance-optimal assignment. AMRIV pairs (i) an online policy that adaptively approximates the optimal allocation with (ii) a sequential, influence-function–based estimator that attains the semiparametric efficiency bound while retaining multiply-robust consistency. We establish asymptotic normality, explicit convergence rates, and anytime-valid asymptotic confidence sequences that enable sequential inference. Finally, we demonstrate the practical effectiveness of our approach through empirical studies, showing that adaptive instrument assignment, when combined with the AMRIV estimator, yields improved efficiency and robustness compared to existing baselines.

## 1 Introduction

Adaptive experimentation enables efficient estimation of treatment effects in sequential settings by adjusting assignment strategies based on accumulating data. Compared to traditional randomized controlled trials (RCTs), adaptive designs can reduce estimation variance, thus accelerating discovery and limiting exposure to ineffective or harmful interventions. These methods are now widely used across domains–from medicine to online platforms–and have been formally endorsed by the U.S. Food and Drug Administration [21], driving both practical adoption and theoretical advances.

In many such settings, however, **direct treatment assignment is not feasible or ethical**. The treatment must instead be *encouraged* through a randomized recommendation or design choice—often referred to as an instrumental variable (IV)—leaving the final decision to the participant. This gives rise to *noncompliance*, where the assigned encouragement and the actual treatment may differ, and where self-selection based on unobserved factors introduces confounding that biases standard estimators. For instance, in a real TripAdvisor experiment, users were exposed to different premium sign-up interfaces that encouraged membership enrollment [52]. The actual treatment—whether a user subscribed—could not be enforced, but the interface (the instrument) could be randomized or adaptively adjusted. Similarly, in clinical trials, a physician may recommend a new medication but cannot compel adherence: the recommendation can be assigned, yet treatment uptake remains endogenous. Related challenges arise in recommender systems and public health interventions, where engagement or behavioral uptake is voluntary.

In such applications, reducing estimator variance is not merely a statistical preference; it determines how quickly and confidently a study can reach conclusions. Because high variance delays both

the detection of harmful effects and the confirmation of beneficial ones, adaptive designs that learn to allocate instruments in variance-minimizing ways can mitigate these risks, yielding tighter confidence sequences and enabling earlier, statistically valid stopping decisions. Despite extensive work on adaptive designs for settings where treatment can be directly enforced [13, 23, 27], adaptive experimentation under noncompliance—where treatment is voluntary but encouragement can be adaptively controlled, remains largely unexplored, even though it describes many real-world scenarios.

This paper addresses this gap. We study the problem of estimating the average treatment effect (ATE) in a sequential experiment where the experimenter can assign only a binary instrument, while the treatment itself is determined endogenously. Building on the semiparametric framework of Wang and Tchetgen Tchetgen [54], which identifies the ATE under an unconfounded compliance assumption and provides a multiply robust, efficient influence-function–based estimator, we extend this framework to the adaptive setting. Specifically:

- We derive the **semiparametric efficiency bound** and characterize the **variance-optimal adaptive policy** that minimizes this bound through a covariate-dependent instrument assignment.

- **We introduce AMRIV**, an **A**daptive, **M**ultiply **R**obust estimator for **IV** settings, which applies a sequential, plug-in version of the efficient influence function evaluated under the adaptive policy.

- We establish **strong theoretical guarantees**, including asymptotic normality, explicit convergence rates, multiply robust consistency, and time-uniform asymptotic confidence sequences for valid inference at arbitrary stopping times.

- We demonstrate **practical effectiveness** in both synthetic and semi-synthetic studies, showing improved efficiency and robustness over non-adaptive baselines and alternatives.

In contrast to prior work on adaptive design with instruments [3, 9, 22, 61], our method focuses on point estimation of the ATE, achieves semiparametric efficiency, and supports multiply robust inference under adaptive assignment. To our knowledge, this is the first method to bring the full suite of modern semiparametric tools—efficient influence functions, adaptive policy learning, robust plug-in estimation, and anytime-valid inference—to the adaptive IV setting with noncompliance.

## 2 Related Work

We provide a brief overview of related work here, with a more detailed discussion in Appendix A.

**IV-Based ATE Estimation.** ATE identification in IV settings has traditionally relied on structural equation models (SEMs) that impose parametric assumptions on the outcome and treatment assignment mechanisms. More recent work has proposed flexible alternatives—such as DeepIV [24], kernel IV [50], and orthogonal moment methods [6, 52]—that enable conditional effect estimation in high-dimensional or nonlinear settings. However, these approaches do not directly target robustness or semiparametric efficiency for the ATE. We instead build on the framework of Wang and Tchetgen Tchetgen [54], which establishes point identification of the ATE via an unconfounded compliance assumption without requiring SEMs. Their influence-function–based estimator achieves semiparametric efficiency and is multiply robust, remaining consistent under partial nuisance misspecification. We extend this framework to the adaptive setting and use it as the foundation for our estimator.

**Adaptive Experimentation for ATE Estimation.** A large and growing literature studies adaptive designs where the treatment itself can be directly assigned, with the goal of minimizing estimator variance or its regret analogue. Early two-stage designs asymptotically achieve the semiparametric efficiency bound [23], and fully sequential approaches such as A2IPW attain variance-optimal Neyman allocation [27]. Subsequent extensions learn the allocation policy online [28] and add principled policy truncation with the first anytime-valid confidence sequences [13]. Parallel work from an online-learning perspective achieves sublinear or logarithmic "Neyman regret" via clipped or optimistic algorithms [14, 38, 39], matches finite-sample lower bounds under low-switching policies [35], and extends to covariate-adaptive and off-policy settings [29, 33]. Together, these methods form a mature toolkit—adaptive nuisance learning, cross-fitting, policy truncation, regret-style allocation, and time-uniform inference—but all assume the experimenter can randomize the *treatment* directly. Our work generalizes these efficiency guarantees to the harder and less explored regime where only an *instrument* can be assigned and treatment uptake is endogenous.

**Adaptive Experimentation with Instruments.** A small but growing literature explores adaptive design in settings where only an instrument, rather than the treatment, can be assigned. Closest to our

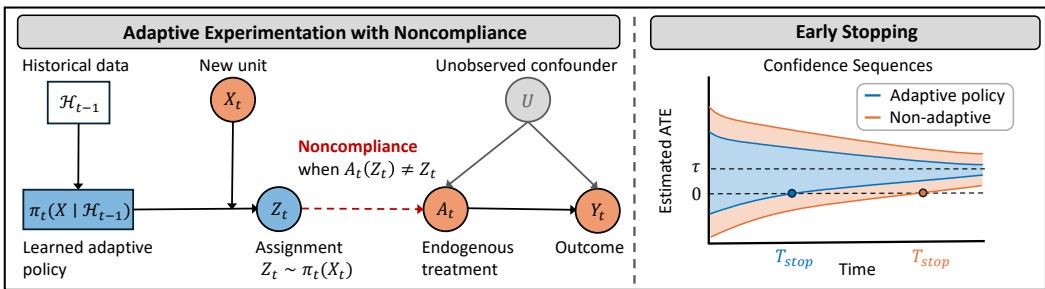

Figure 1: *Left:* Adaptive experimentation with noncompliance. Blue elements denote learned or assigned quantities $(\pi_t, Z_t)$, orange elements represent observed variables $(X_t, A_t, Y_t)$, and the dashed red arrow indicates noncompliance $(A_t(Z_t) \neq Z_t)$. *Right:* The adaptive policy yields faster confidence-sequence contraction, enabling earlier stopping.

work is Chandak et al. [9], who propose a practical influence-function–based procedure to reduce prediction error in nonparametric IV regression. However, they focus on prediction accuracy and do not address semiparametric efficiency or robustness to nuisances. Other approaches focus on partial identification [3], adaptive data acquisition [22], or regret minimization in bandit-style settings with endogeneity [16, 61]. In contrast, our goal is to enable efficient and robust adaptive ATE estimation under noncompliance, providing the first efficient and multiply robust estimator for this setting.

Additional related work on semiparametric inference, multiply robust estimation, and confidence sequences is discussed in detail in Appendix A.

## 3 Background and Setup

We consider the problem of estimating the average treatment effect (ATE) of a binary treatment $A \in \{0, 1\}$ on a real-valued outcome $Y \in \mathbb{R}$, in the presence of unobserved confounding, within an adaptive experimental setting. We adopt the potential outcomes framework, where each unit is associated with two potential outcomes, $Y(0)$ and $Y(1)$, corresponding to the outcomes under control and treatment, respectively. However, only the realized outcome $Y = Y(A)$, corresponding to the treatment actually received, is observed.

Each unit is also associated with covariates $X \in \mathbb{R}^m$, and we assume that the random variables $(X, Y(0), Y(1))$ are jointly distributed according to some unknown distribution $P$. Our goal is to estimate the ATE given by:

$$\tau := \mathbb{E}[Y(1)] - \mathbb{E}[Y(0)].$$

Because direct treatment assignment may be infeasible, we rely on a binary instrumental variable $Z \in \{0, 1\}$ that influences treatment uptake. The instrument can be interpreted as a recommendation or encouragement—something the experimenter can assign, unlike the treatment itself. We denote by $Y(a, z)$ the potential outcome under treatment level $a$ and instrument value $z$, and by $A(z)$ the potential treatment that would be taken under instrument assignment $z$. *Noncompliance* arises when the potential treatment does not follow the instrument, i.e., $A(z) \neq z$, reflecting endogenous treatment selection. Equivalently, a unit complies if its treatment adheres to the assigned instrument $(A(z) = z)$.

The experiment proceeds over $T \in \mathbb{N}$ rounds. At each round $t$, a new unit with covariates $X_t$ is drawn from $P$. The experimenter observes $X_t$ and selects an instrument value $Z_t \sim \pi_t(\cdot \mid X_t, \mathcal{H}_{t-1})$, where $\pi_t$ is an **adaptive policy** that depends on the current covariates $X_t$ and past observations

$$\mathcal{H}_{t-1} := \{(X_1, Z_1, A_1, Y_1), \ldots, (X_{t-1}, Z_{t-1}, A_{t-1}, Y_{t-1})\}.$$

This allows the instrument-assignment policy to evolve over time based on accumulated data. Following the instrument assignment $Z_t$, the treatment $A_t = A_t(Z_t)$ is realized, and the outcome $Y_t = Y(A_t, Z_t)$ is observed. The full observation at time $t$ is thus $(X_t, Z_t, A_t, Y_t)$. After $T$ rounds, the experimenter estimates the ATE from accumulated data $\mathcal{H}_T = \{(X_i, Z_i, A_i, Y_i)\}_{i=1}^{T}$.

To identify the ATE under endogenous treatment selection (that may be influenced by unobserved confounders), we adopt standard instrumental variable assumptions, as well as the unconfounded compliance condition introduced in Wang and Tchetgen Tchetgen [54]. We summarize these below.

**Assumption 1** (Standard IV Assumptions)**.** *The following properties hold: (Exclusion)* $Y(a, z) = Y(a)$—*the instrument affects the outcome only through the treatment; (Independence)* $Z \perp\!\!\!\perp U \mid X$—*the instrument is independent of unobserved confounders* $U$ *given covariates; and (Relevance)* $\mathrm{Cov}(Z, A \mid X) \neq 0$—*the instrument has an effect on treatment uptake for almost every* $X$.

**Assumption 2** (Unconfounded Compliance, from [54])**.** *The treatment effect is independent of compliance status given covariates:* $Y(1) - Y(0) \perp\!\!\!\perp A(1) - A(0) \mid X$.

Assumption 1 is standard in the IV literature and ensures instrument validity. The independence assumption can often be satisfied by design, e.g., via randomization. While sufficient for identifying the local average treatment effect (LATE), these assumptions do not identify the ATE under effect heterogeneity or treatment endogeneity. To enable ATE identification, we invoke Assumption 2 from Wang and Tchetgen Tchetgen [54], which assumes the treatment effect is mean-independent of compliance type given covariates, ruling out interactions with unobserved confounding.

**Remark 1** (Interpretation under violations of Assumption 2)**.** *If Assumption 2 does not hold, the ATE is no longer point-identified. The estimand then shifts to the average conditional local average treatment effect (ACLATE), which averages treatment effects among instrument-responsive individuals (compliers) across covariate strata. The ACLATE remains identified and interpretable, capturing how causal effects vary among compliers when compliance is confounded and cannot be fully controlled.*

With Assumption 1 and 2, the ATE can be point-identified. For notational convenience, we define the instrument-induced outcome and treatment models $\mu^Y(z, X) := \mathbb{E}[Y \mid Z = z, X]$ and $\mu^A(z, X) := \mathbb{E}[A \mid Z = z, X]$ for $z \in \{0, 1\}$. The ATE can then be expressed as (Theorem 1 from [54]):

$$\tau = \mathbb{E}_X \left[ \frac{\mu^Y(1, X) - \mu^Y(0, X)}{\mu^A(1, X) - \mu^A(0, X)} \right] := \mathbb{E}_X \left[ \frac{\delta^Y(X)}{\delta^A(X)} \right], \tag{1}$$

where $\delta^Y(X)$ and $\delta^A(X)$ denote the instrument-induced shifts in outcome and treatment, respectively. We refer to $\delta^A(X)$ as the *compliance score*, representing the instrument's effect on treatment uptake.

In the non-adaptive setting, where the instrument assignment policy is fixed over time—*i.e.*, $\pi_t(1 \mid X, \mathcal{H}_{t-1}) \equiv \pi(X)$—Wang and Tchetgen Tchetgen [54] (Theorem 5) derive the efficient influence function (EIF) for the ATE estimator in Equation 1. Let $\pi(x), \eta(x) := \{\mu^Y(0, x), \mu^A(0, x), \delta^A(x), \delta(x)\}$ denote the nuisances, where $\delta(X) := \delta^Y(X)/\delta^A(X)$. The (Recentered) EIF is then given by

$$\phi(X, Z, A, Y; \pi, \eta) \tag{2}$$
$$= \frac{2Z - 1}{Z\pi(X) + (1 - Z)(1 - \pi(X))} \frac{1}{\delta^A(X)} \left[ Y - A\delta(X) - \mu^Y(0, X) + \mu^A(0, X)\delta(X) \right] + \delta(X).$$

The corresponding estimator—known as the multiply robust IV estimator (MRIV) [18, 54]—uses plug-in estimates of nuisance functions within the recentered efficient influence function. It attains the semiparametric efficiency bound when all nuisances are correctly specified and remains consistent under partial misspecification.

We extend this framework to the adaptive setting, where the instrument assignment policy $\pi_t$ evolves with accumulating data. We characterize the optimal adaptive policy that minimizes asymptotic variance and develop **AMRIV**, an **A**daptive **M**ultiply **R**obust **IV** estimator that combines adaptive policy learning with sequential influence-function–based estimation. Our method achieves semiparametric efficiency, ensures multiply robust consistency, and enables valid time-uniform inference.

**Notation:** We write $\pi_t(X_t \mid \mathcal{H}_{t-1}) := \pi_t(1 \mid X_t, \mathcal{H}_{t-1})$ for the probability of assigning $Z_t = 1$ given covariates and history. The $L_2$ norm of a function $f$ is $\|f\|_2 := \mathbb{E}_P[f(X)^2]^{1/2}$, and $\widehat{f}_t$ denotes an estimate of $f$ based on $t$ samples. We use $\widehat{\mathbb{E}}$ to denote empirical expectations computed from data. Notation used throughout the paper is summarized in Appendix B (Table 1).

## 4 Efficiency Bounds and Optimal Instrument Assignment

To guide optimal experiment design under the IV setting, we derive the semiparametric efficiency bound for ATE estimation under a fixed instrument policy $\pi(X)$. This characterizes the variance-minimizing allocation strategy and motivates our adaptive estimator.

**Theorem 1** (Semiparametric Efficiency Bound). *Under Assumption 1 and 2, the semiparametric efficiency bound for estimating the ATE $\tau$ is given by*

$$V_{\mathit{eff}}(\pi) := \mathbb{E}\left[\frac{1}{\delta^A(X)^2}\left(\frac{\sigma^2(1, X)}{\pi(X)} + \frac{\sigma^2(0, X)}{1 - \pi(X)}\right) + (\delta(X) - \tau)^2\right], \tag{3}$$

*where $\sigma^2(z, X) = \mathrm{Var}(Y - A\delta(X) \mid Z = z, X)$.*

**Corollary 2** (Optimal Instrument Assignment). *The assignment policy $\pi^*(X)$ that minimizes the efficiency bound in Theorem 1 is given by*

$$\pi^*(X) = \frac{\sqrt{\sigma^2(1, X)}}{\sqrt{\sigma^2(1, X)} + \sqrt{\sigma^2(0, X)}}. \tag{4}$$

Proofs of Theorem 1 and Corollary 2 are included in Appendix E.

**Drivers of Efficient Allocation.** From Corollary 2, the optimal assignment policy $\pi^*(X)$ allocates more weight to the arm $z$ with higher residual variance $\mathrm{Var}(Y - A\delta(X) \mid Z = z, X)$, where

$$\mathrm{Var}(Y - A\delta(X) \mid Z = z, X) = \mathrm{Var}(Y \mid Z = z, X) + \delta(X)^2 \, \mathrm{Var}(A \mid Z = z, X) \\ - 2\delta(X) \, \mathrm{Cov}(Y, A \mid Z = z, X).$$

This expression reveals that, unlike in standard adaptive ATE estimation, the residual variance depends jointly on both outcome noise ($\mathrm{Var}(Y \mid Z = z, X)$) and compliance noise ($\mathrm{Var}(A \mid Z = z, X)$). When compliance is more uncertain in one arm, the estimator becomes noisier in that region, leading the optimal policy to allocate more probability mass to that arm to compensate.

**Connection to Neyman Allocation.** In the special case of perfect compliance (when $A(Z) = Z$), the treatment is fully determined by the (conditionally) randomized instrument and our setting becomes the classical adaptive ATE estimation scenario. In this setting, $\mathrm{Var}(Y - A\delta(X) \mid Z = z, X) = \mathrm{Var}(Y - A\delta(X) \mid A = z, X) = \mathrm{Var}(Y \mid A = z, X)$ and thus the optimal allocation reduces to $\frac{\sqrt{\mathrm{Var}(Y \mid A=1, X)}}{\sqrt{\mathrm{Var}(Y \mid A=0, X)} + \sqrt{\mathrm{Var}(Y \mid A=1, X)}}$ which exactly matches the classical Neyman allocation for minimizing the variance of a difference-in-means estimator [27, 40]. This highlights that our policy generalizes Neyman allocation to settings with noncompliance and endogenous treatment, adjusting for both outcome and compliance-driven noise.

**Motivating Illustration.** Consider an example with *one-sided noncompliance*—treatment is only accessible to those who receive the instrument, so $\mu^A(0, X) = 0$. This captures scenarios such as vaccine access, product rollouts, or behavioral nudges. Let compliance vary with $X \in \mathbb{R}$ via $\delta^A(x) = \mu^A(1, x) = \sigma(-2x)$, and let outcomes follow $Y = f(A, X) + uA + \epsilon_A$, where $u$ is a fixed unobserved confounder and $\epsilon_A \sim \mathcal{N}(0, A + 4(1 - A))$, with higher variance in the control arm. As shown in Figure 2, the optimal policy $\pi^*(X)$ approaches Neyman allocation when $\delta^A(X) \to 1$, but shifts toward uniform allocation when $\delta^A(X) \to 0$. This reflects a key design intuition: *under low compliance, assigning more units to $Z = 1$ compensates for scarce treatment uptake*, helping preserve estimator efficiency.

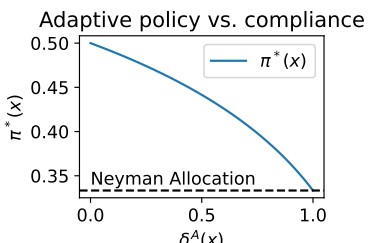

Figure 2: Optimal policy $\pi^*(X)$ as a function of compliance $\delta^A(X)$.

## 5 Adaptive Estimation of the Average Treatment Effect

We propose an adaptive framework for estimating the ATE in sequential experiments with a binary instrument. Our goal is to minimize the semiparametric efficiency bound from Theorem 1 by combining: (1) **instrument assignment** via a data-driven policy $\pi_t(X_t \mid \mathcal{H}_{t-1})$ that approximates the optimal allocation $\pi^*(X)$; and (2) **treatment effect estimation** using an adaptive plug-in version of the multiply robust estimator from Equation 2. Although the theory assumes per-round updates, the method also applies in batch settings with fewer updates. We detail both components below.

## 5.1 Adaptive Instrument Assignment

To stabilize nuisance estimation, we begin with a burn-in phase of $T_0 < T$ rounds using a fixed policy $\pi_{\text{init}}(X)$, such as uniform randomization. From round $T_0 + 1$ onward, instruments are assigned via a data-driven policy $\pi_t(X \mid \mathcal{H}_{t-1})$ that approximates the optimal allocation in Corollary 2. Specifically, we compute a plug-in estimate $\widetilde{\pi}_t(X \mid \mathcal{H}_{t-1})$ as

$$\widetilde{\pi}_t(X \mid \mathcal{H}_{t-1}) := \frac{\sqrt{\widehat{\sigma}_{t-1}^2(1, X)}}{\sqrt{\widehat{\sigma}_{t-1}^2(0, X)} + \sqrt{\widehat{\sigma}_{t-1}^2(1, X)}}, \tag{5}$$

where $\widehat{\sigma}_{t-1}^2(z, X)$ is an estimate of the conditional residual variance $\text{Var}(Y - A\delta(X) \mid Z = z, X)$ based on data in $\mathcal{H}_{t-1}$. We then apply a truncation step to $\widetilde{\pi}_t(X \mid \mathcal{H}_{t-1})$ (described below) to obtain the final assignment policy $\pi_t(X \mid \mathcal{H}_{t-1})$ used at time $t$.

**Residual–variance estimation.** One option for estimating $\widehat{\sigma}_{t-1}^2(z, X)$ is using the decomposition

$$\text{Var}(Y - A\delta(X) \mid Z = z, X) = \mathbb{E}[(Y - A\delta(X))^2 \mid Z = z, X] - \left(\mu^Y(0, X) - \mu^A(0, X)\,\delta(X)\right)^2.$$

We proceed in two stages: (i) fit $\widehat{\mu}_{t-1}^Y(0, X)$, $\widehat{\mu}_{t-1}^A(0, X)$ and $\widehat{\delta}_{t-1}(X)$ using $\mathcal{H}_{t-1}$; (ii) form residuals $\widehat{R}_{t-1} = Y - A\widehat{\delta}_{t-1}(X)$ and regress $\widehat{R}_{t-1}^2$ on $(Z, X)$ to obtain $\widehat{s}_{t-1}(z, X) := \widehat{\mathbb{E}}[\widehat{R}_{t-1}^2 \mid Z = z, X]$.

**Unbiased two-stage estimation via cross-fitting.** To mitigate finite-sample bias in estimating $\widehat{\sigma}_{t-1}^2(z, X)$, we apply the sequential cross-fitting scheme of Waudby-Smith et al. [55]. Thus, we split $\mathcal{H}_{t-1}$ into two temporal folds $\mathcal{H}_{t-1}^{(j)} = \{(X_i, Z_i, A_i, Y_i) : i \in [t-1], i \mod 2 = j\}$, $j \in \{0, 1\}$, fit $\widehat{\delta}_{t-1}$ on one and compute residuals $\widehat{R}_{t-1}^2$ on the other, and *vice-versa*. The combined residuals are used to regress $\widehat{R}_{t-1}^2$ on $(Z, X)$ to estimate $\widehat{s}_{t-1}(z, X)$. Since $\widehat{\mu}_{t-1}^Y$ and $\widehat{\mu}_{t-1}^A$ do not depend on other nuisances, they can be learned on the full history $\mathcal{H}_{t-1}$.

**Nuisance learners.** Any sequentially consistent nonparametric regressor can be used for $\widehat{\mu}_{t-1}^Y, \widehat{\mu}_{t-1}^A$, and $\widehat{s}_{t-1}$, *e.g.* $k$-NN [58], kernel smoothers [45], random forests [53], or neural nets [48]. $\widehat{\mu}_{t-1}^A$ may also be estimated via these methods or a classifier such as logistic regression. We compute $\widehat{\delta}_{t-1}(X) = \widehat{\delta}_{t-1}^Y(X)/\widehat{\delta}_{t-1}^A(X)$, where $\widehat{\delta}_{t-1}^Y$ is estimated via either a difference of regressions $\widehat{\mu}_{t-1}^Y(1, X) - \widehat{\mu}_{t-1}^Y(0, X)$ or a direct IPW-style regression $\widehat{\mathbb{E}}\left[\frac{YZ}{\pi_{t-1}(X)} - \frac{Y(1-Z)}{1-\pi_{t-1}(X)} \mid X\right]$. An estimate of $\widehat{\delta}_{t-1}^A(X)$ is obtained analogously by replacing $Y$ with $A$.

To guarantee non-negativity of the estimated variances, we define $\widehat{\sigma}_{t-1}^2(z, X)$ as

$$\widehat{\sigma}_{t-1}^2(z, X) = \begin{cases} \left\{\widehat{s}_{t-1}(z, X) - (\widehat{\mu}_{t-1}^Y(0, X) - \widehat{\mu}_{t-1}^A(0, X)\,\widehat{\delta}_{t-1}(X))^2\right\} & \text{if } \{\cdots\} > 0 \\ \varepsilon & \text{otherwise} \end{cases} \tag{6}$$

for a small constant $\varepsilon > 0$.

**Remark 2** (Choice of nuisance and variance estimators). *Sequential cross-fitting removes the need for restrictive conditions (e.g., Donsker)—standard nonparametric convergence rates suffice for the guarantees in our main theorems. In practice, any consistent learner (e.g., $k$-NN, random forests, or neural networks) can be used, depending on data structure and sample size (see Appendix C). The variance estimator in Eq. (6) ensures nonnegativity via a small floor $\varepsilon$, though fully nonnegative alternatives such as self-normalized kernel estimators may also be used. Appendix C further discusses these options and outlines online or streaming implementations that avoid full data storage.*

**Truncation for Finite-Sample Stability.** Following recent work on adaptive ATE estimation without endogenous treatment assignment (*e.g.*, [13, 14, 38]), we apply a truncation scheme to stabilize the assignment policy $\widetilde{\pi}_t(X \mid \mathcal{H}_{t-1})$. After computing the raw plug-in policy $\widetilde{\pi}_t(X \mid \mathcal{H}_{t-1})$ via Eq. (5), we define the truncated policy $\pi_t(X \mid \mathcal{H}_{t-1})$ as

$$\pi_t(X \mid \mathcal{H}_{t-1}) := \min\left\{1 - \frac{1}{k_t}, \max\left\{\frac{1}{k_t}, \widetilde{\pi}_t(X \mid \mathcal{H}_{t-1})\right\}\right\}, \tag{7}$$

where $k_t \in [2, \infty)$ is a truncation parameter satisfying $k_t \to \infty$ as $t \to \infty$. Truncation ensures that the instrument assignment probabilities remain bounded away from 0 and 1, thereby improving finite-sample stability and leading to better theoretical guarantees.

---

**Algorithm 1** AMRIV: Adaptive Multiply Robust IV Estimation

---

**Require:** Burn-in period $T_0$; initial policy $\pi_{\text{init}}(X)$; regression/classification learners for $\mu^Y(z, X)$, $\mu^A(z, X)$, $\delta(X)$, $\delta^A(X)$, $s(z, X)$.

1: **for** $t = 1$ **to** $T$ **do**
2:     Observe covariates $X_t$.
3:     **if** $t \leq T_0$ **then**
4:         Assign $Z_t \sim \text{Bern}(\pi_{\text{init}}(X_t))$.
5:     **else**
6:         Estimate nuisance functions $\widehat{\mu}^Y_{t-1}(0, X)$, $\widehat{\mu}^A_{t-1}(0, X)$, $\widehat{\delta}_{t-1}(X)$, and $\widehat{s}_{t-1}(z, X)$ from $\mathcal{H}_{t-1}$ using cross-fitting. Compute $\widehat{\sigma}^2_{t-1}(z, X)$ using Eq. (6).
7:         Compute plug-in assignment probability: $\widetilde{\pi}_t(X \mid \mathcal{H}_{t-1}) = \frac{\sqrt{\widehat{\sigma}^2_{t-1}(1,X)}}{\sqrt{\widehat{\sigma}^2_{t-1}(0,X)} + \sqrt{\widehat{\sigma}^2_{t-1}(1,X)}}$.
8:         Apply truncation to obtain $\pi_t(X \mid \mathcal{H}_{t-1}) := \min\left\{1 - \frac{1}{k_t}, \max\left\{\frac{1}{k_t}, \widetilde{\pi}_t(X \mid \mathcal{H}_{t-1})\right\}\right\}$.
9:         Assign $Z_t \sim \text{Bern}(\pi_t(X_t \mid \mathcal{H}_{t-1}))$.
10:    Observe instrumented treatment $A_t = A(Z_t)$ and outcome $Y_t = Y(A_t)$.
11:    Construct $\widehat{\eta}_t = \{\widehat{\mu}^Y_{t-1}(0, X), \widehat{\mu}^A_{t-1}(0, X), \widehat{\delta}^A_{t-1}(X), \widehat{\delta}_{t-1}(X)\}$ by estimating (or reusing) nuisance functions via cross-fitting.
12:    Compute $\phi_t = \phi(X_t, Z_t, A_t, Y_t; \pi_t, \widehat{\eta}_t)$ using Eq. (9).
13: **return** $\widehat{\tau}^{\text{AMRIV}}_T = \frac{1}{T}\sum_{t=1}^{T} \phi_t$.

---

## 5.2 AMRIV: Adaptive Multiply Robust Estimation of the ATE

We now introduce our estimator, **AMRIV**, which adaptively estimates the ATE using the recentered efficient influence function in Eq. (2) evaluated on sequentially updated plug-in estimates of nuisance functions. The estimator is defined as

$$\widehat{\tau}^{\text{AMRIV}}_T := \frac{1}{T}\sum_{t=1}^{T} \phi(X_t, Z_t, A_t, Y_t; \pi_t, \widehat{\eta}_t), \tag{8}$$

where $\widehat{\eta}_t = \{\widehat{\mu}^Y_{t-1}(0, X), \widehat{\mu}^A_{t-1}(0, X), \widehat{\delta}^A_{t-1}(X), \widehat{\delta}_{t-1}(X)\}$ denotes plug-in estimates of the nuisance functions at time $t$, constructed solely from the past data $\mathcal{H}_{t-1}$ (Note: the instrument assignment policy $\pi_t(X \mid \mathcal{H}_{t-1})$, defined by the experimenter based on the estimated optimal rule from Section 5.1, is treated as known and does not require further estimation from data). This construction confers the estimator $\widehat{\tau}^{\text{AMRIV}}_T$ a *near-martingale* structure, that is, it can be written as the sum of a true martingale difference sequence and a remainder term of order $o_p(T^{-1/2})$, enabling, as we will show in Section 6, valid asymptotic inference under sequential dependence.

**Nuisance Estimation.** The nuisance functions $\widehat{\mu}^Y_{t-1}(0, X), \widehat{\mu}^A_{t-1}(0, X), \widehat{\delta}^A_{t-1}(X), \widehat{\delta}_{t-1}(X)$ can be estimated using any flexible nonparametric regression method applied to the historical data $\mathcal{H}_{t-1}$. To reduce computational overhead, we can reuse the estimates of $\widehat{\mu}^Y_{t-1}(0, X)$ and $\widehat{\mu}^A_{t-1}(0, X)$ previously obtained for instrument assignment in Section 5.1. Similarly, the estimate of $\widehat{\delta}_{t-1}(X)$ can be formed by averaging the cross-fitted estimates $\widehat{\delta}^{(0)}_{t-1}$ and $\widehat{\delta}^{(1)}_{t-1}$, trained on the two data folds $\mathcal{H}^{(0)}_{t-1}$ and $\mathcal{H}^{(1)}_{t-1}$, respectively. The only remaining component is $\widehat{\delta}^A_{t-1}(X)$, which must be estimated separately if it was not already computed as part of the $\widehat{\delta}_{t-1}(X)$ estimation pipeline.

For completeness, the final estimate of the (R)EIF at time $t$ is given by

$$\phi(X_t, Z_t, A_t, Y_t; \pi_t, \widehat{\eta}_t) = \frac{2Z_t - 1}{Z_t\pi_t(X_t \mid \mathcal{H}_{t-1}) + (1 - Z_t)(1 - \pi_t(X_t \mid \mathcal{H}_{t-1}))} \frac{1}{\widehat{\delta}^A_{t-1}(X_t)}$$
$$\cdot \left[Y_t - A_t\widehat{\delta}_{t-1}(X_t) - \widehat{\mu}^Y_{t-1}(0, X_t) + \widehat{\mu}^A_{t-1}(0, X_t)\widehat{\delta}_{t-1}(X_t)\right] + \widehat{\delta}_{t-1}(X_t) \tag{9}$$

where all quantities are constructed from $\mathcal{H}_{t-1}$. The full procedure is summarized in Algorithm 1. Unlike prior adaptive ATE methods without IVs, the estimator $\widehat{\tau}^{\text{AMRIV}}_T = \frac{1}{T}\sum_{t=1}^{T} \phi(X_t, Z_t, A_t, Y_t; \pi_t, \widehat{\eta}_t)$ *cannot* be written as a martingale difference sequence. Hence, standard MDS central limit theorems do not apply directly, and we must instead decompose the estimator to recover a suitable martingale structure.

# 6 Theoretical Guarantees

This section provides theoretical guarantees for the AMRIV estimator. We establish its asymptotic normality, characterize its convergence rates, and demonstrate its multiply-robust consistency. Furthermore, in Appendix D, we consider the sequential inference setting and derive asymptotically-valid, time-uniform *confidence sequences* for the AMRIV estimator.

## 6.1 Efficiency and Asymptotic Normality of the AMRIV Estimator

We start by establishing the asymptotic properties of the AMRIV estimator $\widehat{\tau}_T^{\text{AMRIV}}$. We first introduce the following assumption:

**Assumption 3** (Bounded Outcomes and Nuisances). *The potential outcomes and nuisance function estimates are uniformly bounded. That is, there exists a constant $C > 0$ such that, for all $t$ and $x$:* $|Y_t(0)|, |Y_t(1)| \leq C, |\widehat{\mu}_t^Y(0, x)|, |\widehat{\mu}_t^A(0, x)|, |\widehat{\delta}_t(x)|, |\widehat{\delta}_t^A(x)|^{-1} \leq C.$

This boundedness assumption is standard in influence-function-based ATE estimation and ensures stability of the estimator. With this assumption in place, we now state our main result on the asymptotic efficiency of the AMRIV estimator.

**Theorem 3** (Asymptotic Normality of the AMRIV Estimator). *Suppose Assumptions 1 to 3 hold and there exists a non-adaptive policy $\pi(X) \in [\epsilon, 1-\epsilon]$ for some $\epsilon > 0$ such that the nuisances estimates $\widehat{\eta}_t$ and the adaptive assignment policy $\pi_t(X \mid \mathcal{H}_{t-1})$ are $L_2$-consistent relative to the truncation schedule, i.e. $k_t\|\widehat{f}_{t-1} - f\|_2 = o_p(1)$ and $k_t\|\pi_t - \pi\|_2 = o_p(1)$ for $f \in \{\mu^Y(0, \cdot), \mu^A(0, \cdot), \delta(\cdot), \delta^A(\cdot)\}$. Furthermore, assume $\|\widehat{\delta}_{t-1} - \delta\|_2\|\widehat{\delta}_{t-1}^A - \delta^A\|_2 = o_p(t^{-1/2})$. Then, the AMRIV estimator is asymptotically normal:*

$$\sqrt{T}\left(\widehat{\tau}_T^{\text{AMRIV}} - \tau\right) \xrightarrow{d} \mathcal{N}\left(0, V_{\text{eff}}(\pi)\right), \tag{10}$$

*where $V_{\text{eff}}$ is defined in Theorem 1. In particular, if we have $\pi(X) = \pi^*(X)$, then $\widehat{\tau}_T^{\text{AMRIV}}$ is semiparametrically efficient.*

The key insight behind Theorem 3 is that the AMRIV estimator admits the following near-martingale decomposition: $\sqrt{T}(\widehat{\tau}_T^{\text{AMRIV}} - \tau) = \sqrt{T}\left(\frac{1}{T}\sum_{t=1}^T z_t\right) + \sqrt{T}\left(\frac{1}{T}\sum_{t=1}^T m_t\right)$, where $z_t = \phi(X_t, Z_t, A_t, Y_t; \pi_t, \eta) - \tau$ is a martingale difference sequence (MDS), and $m_t = \phi(X_t, Z_t, A_t, Y_t; \pi_t, \widehat{\eta}_t) - \phi(X_t, Z_t, A_t, Y_t; \pi_t, \eta)$ is an asymptotically vanishing term that captures the impact of estimating the nuisance functions. The first term satisfies a central limit theorem for MDS under standard Lindeberg-type conditions [60], while the second is controlled by the $L_2$-consistency of the nuisance estimates. We formalize this in Appendix F. Importantly, this result holds under mild assumptions: we only require $L_2$ convergence (no pointwise convergence or Donsker conditions [7]), bounded outcomes and nuisance estimates, and $L_2$-consistency of the nuisance components w.r.t. the truncation schedule. This allows AMRIV to accommodate flexible, data-dependent policies and sequential nuisance estimation.

The truncation schedule $k_t$ plays a central role by ensuring positivity of $\pi_t(X)$—crucial for variance control—while still allowing $\pi_t$ to approach an optimal policy $\pi^*$ as $k_t \to \infty$. For this to hold, we must ensure $\lim_{t\to\infty} k_t > \sup_X \frac{1}{\pi^*(X)}$, so the truncation threshold does not constrain the optimal allocation in the limit and semiparametric efficiency can be achieved (see last line in Theorem 3). This mirrors tradeoffs in efficient ATE estimation [13], where adaptive truncation stabilizes estimation without distorting the estimator asymptotically. In practice, when the plug-in policy $\tilde{\pi}_t$ is uniformly bounded away from 0 and 1, truncation becomes unnecessary: setting $k_t = 1/\min_X\{\tilde{\pi}_t(X), 1 - \tilde{\pi}_t(X)\}$ ensures $\pi_t = \tilde{\pi}_t$ for all $t$.

## 6.2 Consistency Guarantees under Partial Nuisance Misspecification

As shown in Theorem 3, the convergence rate of AMRIV is primarily governed by the estimation error of $\widehat{\delta}(X)$ and $\widehat{\delta}^A(X)$. This reflects its *multiply robust* property: AMRIV remains consistent even when other nuisance components are misspecified. This robustness goes beyond prior work on IV methods in adaptive settings, where such guarantees were not established. The next two results formalize AMRIV's convergence rate and multiply robust consistency.

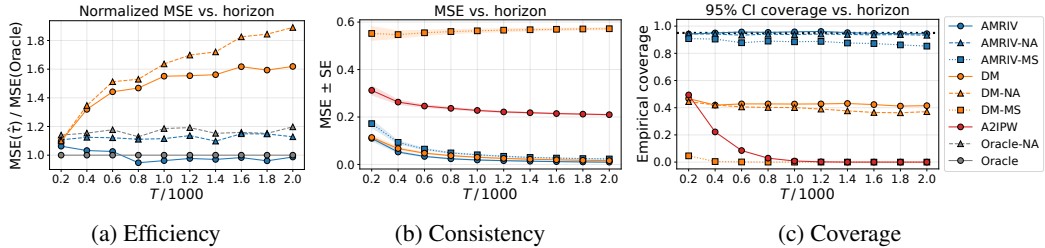

Figure 3: Performance of different estimators across increasing sample size $T$. **(a)** Efficiency: Normalized MSE versus an oracle benchmark. **(b)** Consistency: MSE $\pm$ standard error. **(c)** Coverage: Empirical coverage of 95% confidence intervals.

**Theorem 4** (Convergence Rate of the AMRIV Estimator). *Suppose Assumptions 1 to 3 hold, and that there exists a non-adaptive policy $\pi(X) \in [\epsilon, 1 - \epsilon]$ for some $\epsilon > 0$ such that the adaptive policies $\pi_t$ satisfy $k_t \|\pi_t - \pi\|_2 = o_p(1)$. Let $\widetilde{\eta} = \{\widetilde{\mu}^Y(0, \cdot), \widetilde{\mu}^A(0, \cdot), \widetilde{\delta}(\cdot), \widetilde{\delta}^A(\cdot)\}$ denote a possibly misspecified limit of the nuisance functions, and suppose that $k_t \|\widehat{f}_{t-1} - \widetilde{f}\|_2 = o_p(1)$ for each $\widetilde{f} \in \widetilde{\eta}$. Then the AMRIV estimator satisfies*

$$\left|\widehat{\tau}_T^{\mathrm{AMRIV}} - \tau\right| = O_p(T^{-1/2}) + O_p\left(\|\widehat{\delta}_T^A - \delta^A\|_2 \|\widehat{\delta}_T - \delta\|_2\right). \tag{11}$$

**Corollary 5** (Multiply Robust Consistency Guarantees). *Under the conditions of Theorem 4, if either $\widehat{\delta}_t$ or $\widehat{\delta}_t^A$ is $L_2$-consistent, then the AMRIV estimator $\widehat{\tau}_T^{\mathrm{AMRIV}}$ is consistent for $\tau$.*

We provide a proof of Theorem 4 and Corollary 5 in Appendix G. An immediate consequence of Theorem 4 is that if both $\widehat{\delta}(X)$ and $\widehat{\delta}^A(X)$ converge at rate $o_p(T^{-1/4})$, AMRIV achieves the parametric $O_p(T^{-1/2})$ rate. This is usually attainable under mild regularity conditions, even with flexible nonparametric models. Furthermore, Corollary 5 shows that AMRIV inherits the *multiply robust* property from its static counterpart [18, 54]. In the static setting, the MRIV converges if either (i) $\widehat{\mu}^Y(0, \cdot)$, $\widehat{\mu}^A(0, \cdot)$, $\widehat{\delta}$ are correctly specified, (ii) both $\widehat{\pi}$ and $\widehat{\delta}^A$ are, or (iii) $\widehat{\pi}$ and $\widehat{\delta}$ are. However, in the adaptive setting, we can establish a stronger result: even if the outcome-related nuisance functions $\widehat{\mu}_t^Y(0, \cdot)$ and $\widehat{\mu}_t^A(0, \cdot)$ are misspecified, AMRIV is still consistent as long as *one* of $\widehat{\delta}_t^A$ or $\widehat{\delta}_t$ converges. This is due to the adaptive setting design where we control the instrument assignment $\pi_t(X_t \mid \mathcal{H}_{t-1})$ which confers robustness to misspecification in $\widehat{\mu}_t^Y(0, \cdot)$ and $\widehat{\mu}_t^A(0, \cdot)$. Thus, our adaptive generalization preserves the multiple robustness property, making it particularly well-suited for practice where some nuisance components may be difficult to estimate reliably.

## 7 Experimental Results

We demonstrate the practical effectiveness of our approach in both synthetic and semi-synthetic studies. In each setting, we compare our estimator (**AMRIV**) to its non-adaptive counterpart (**AMRIV-NA**), which assigns the instrument uniformly at random; the plug-in direct method from Eq. (1) (**DM**) and its non-adaptive version (**DM-NA**); the **A2IPW** estimator from [27]; and two oracle baselines: a fully oracle-efficient estimator that uses the true nuisance functions (**Oracle**) and a non-adaptive version (**Oracle-NA**). To assess robustness, we also evaluate misspecified variants of AMRIV and DM—denoted **AMRIV-MS** and **DM-MS**—in which the $\delta(X)$ estimator is deliberately misspecified.

Across both experiments, we evaluate three desiderata: (i) *efficiency*, measured by normalized MSE relative to the Oracle estimator; (ii) *consistency*, assessed via MSE decay with sample size $T$; and (iii) *coverage*, computed from empirical 95% confidence intervals. We implement all estimators using Random Forests (RF, [8]) and update the nuisance estimates in mini-batches for efficiency. Further implementation details, including model hyperparameters and ablations, are provided in Appendix H. The replication code is available at `https://github.com/CausalML/Adaptive-IV`.

### 7.1 Simulation Studies with Synthetic Data

We construct a synthetic environment with one-sided noncompliance, where the treatment $A$ is only accessible to those who receive the instrument $Z = 1$. At each time $t$, we sample covariates $X_t$, assign

the instrument $Z_t \sim \pi_t(X_t \mid \mathcal{H}_{t-1})$, and realize $A_t = C_t Z_t$, where $C_t$ is a latent compliance indicator sampled from $\mathrm{Bern}(\delta^A(X_t))$. The outcome $Y_t$ depends on $A_t$, $X_t$, an unobserved confounder $U_t$, and heteroskedastic noise. The full data-generating process is detailed in Appendix H.1.

We set $T = 2000$, $T_0 = 200$, and run 1000 trajectories. All estimators are updated in batches of size $b = 200$ and implemented using Random Forests (RFs) when applicable. For the adaptive estimators, we use the truncated optimal policy in Eq. (7), with truncation schedule $k_t = 2/0.999^t$. AMRIV uses RF classifiers for $\delta^A(X)$ (clipped at 0.01) and RF regressors for $\mu^Y(z, X)$, while $\delta(X)$ is computed via the plug-in ratio. A2IPW follows Kato et al. [27] with Neyman allocation and RF regressors. To induce misspecification, we replace $\widehat{\mu}^Y(1, X)$ with the constant $\widehat{\mathbb{E}}[\mu^Y(1, X)]$, flattening outcome heterogeneity. Figure 3 summarizes the experimental results.

**Adaptivity.** As shown in panel (a), adaptive design consistently improves the efficiency of all estimators. AMRIV approaches the oracle benchmark despite using estimated nuisances, while AMRIV-NA exhibits a constant efficiency gap due to suboptimal allocation. This illustrates how adaptivity enables more effective data collection: by dynamically allocating instruments to regions of high uncertainty, AMRIV concentrates sampling effort where it contributes most to precision. The effect is particularly evident under one-sided noncompliance, where asymmetries in both outcome and compliance variance make uniform allocation especially inefficient (Theorem 1). Panel (a) also confirms that AMRIV and AMRIV-NA converge at the expected $O_p(T^{-1/2})$ rate (Theorem 4), whereas DM and DM-NA converge more slowly, as their normalized MSE increases with $T$.

**Consistency.** Panel (b) confirms that AMRIV, AMRIV-NA, and DM converge to the true $\tau$, with AMRIV variants achieving lower error due to variance-aware allocation. In contrast, A2IPW is biased and fails to converge, as expected, since it does not correct for unobserved confounding in treatment selection. The comparison further highlights the importance of robust nuisance estimation: DM-MS diverges when $\delta(X)$ is misspecified, while AMRIV-MS remains consistent. This robustness reflects the multiply-robust property formalized in Theorem 4, which ensures consistency as long as either the compliance model or the outcome model is estimated correctly—an especially desirable feature in practice when some nuisance components are difficult to learn reliably.

**Coverage.** In panel (c), we evaluate the empirical coverage of 95% asymptotic confidence intervals. Only AMRIV and AMRIV-NA achieve nominal coverage, consistent with our theoretical guarantees (Theorem 3). The misspecified and plug-in methods under-cover, with DM and A2IPW performing particularly poorly as $T$ grows, owing to finite-sample bias and unaddressed confounding bias, respectively. AMRIV-MS provides partial correction but still falls short of nominal coverage, consistent with the requirement that $\delta(X)$ be estimated consistently for asymptotic validity.

### 7.2 Simulation Studies with Semi-Synthetic Data

We also evaluate AMRIV on a semi-synthetic dataset based on the TripAdvisor customer simulator from Syrgkanis et al. [52], where we use customer features as covariates $X$, a simulated signup prompt as the instrument $Z$, and subscription revenue as the outcome $Y$. The DGP and oracle nuisances are described in Appendix H.2. Results are consistent with the synthetic setting: adaptive instrument assignment improves efficiency, AMRIV achieves superior coverage and consistency, and robustness holds under partial misspecification.

Overall, these findings confirm that pairing adaptive design with the AMRIV estimator improves efficiency, enhances robustness, and yields more reliable inference than non-adaptive baselines.

## 8 Conclusion

We develop AMRIV, an adaptive, multiply robust estimator for ATE estimation in experiments where treatment can only be encouraged via a binary instrument. Our approach (i) derives the semiparametric efficiency bound and optimal assignment policy, (ii) constructs a sequential estimator that attains the bound under adaptive allocation, (iii) provides asymptotic normality, convergence rates, and multiply robust consistency, and (iv) supports valid inference through time-uniform confidence sequences. Empirical results on synthetic and semi-synthetic data confirm that adaptive instrument assignment improves both efficiency and robustness over non-adaptive baselines. This work represents a step toward principled, data-efficient experimentation in real-world settings where compliance is optional and uncertainty is unavoidable. We discuss limitations and broader impacts in Appendix I.

**Acknowledgements**

We thank the anonymous reviewers for their thoughtful feedback and insightful suggestions, which have helped improve and strengthen this work. Miruna Oprescu was supported by the U.S. Department of Energy, Office of Science, Office of Advanced Scientific Computing Research, under Award DE-SC0023112. Brian M. Cho was supported by the Department of Defense (DoD) through the National Defense Science & Engineering Graduate (NDSEG) Fellowship Program. Nathan Kallus was supported by the U.S. National Science Foundation under Grant No. 1846210. Any opinions, findings, and conclusions or recommendations expressed in this material are those of the author(s) and do not necessarily reflect the views of the U.S. Department of Energy, the U.S. Department of Defense, or the U.S. National Science Foundation.

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

# A  Extended Literature Review

We contextualize our work by surveying six strands of prior research. We group these into two categories: (1) *core* threads that directly motivate and inform our methodology, and (2) *auxiliary* threads that provide important theoretical and practical foundations but are not specific to our design.

## A.1  Core Related Work

**Identification and Estimation with Instrumental Variables.** Instrumental variable (IV) methods are widely used to estimate causal effects in the presence of endogenous treatment selection due to unmeasured confounding. Under classical IV assumptions—exclusion, independence, and relevance—these methods typically identify the local average treatment effect (LATE) [1, 2, 4, 10, 42], which pertains to compliers: units whose treatment responds to the instrument. However, compliers represent an unknown and potentially unrepresentative subpopulation, limiting the policy relevance of LATE.

To target the population average treatment effect (ATE), IV methods have traditionally relied on linear structural equation models (SEMs), in which the ATE corresponds to a regression coefficient under correct model specification [19]. Two-stage least squares (2SLS) is the canonical estimator in this setting, but its consistency and interpretability depend on strong linearity assumptions [12, 57]. More recent SEM-based IV approaches focus on estimating conditional effects, including kernel IV [50], DeepIV [24], and other moment-based estimators [6, 52].

We instead build on the framework of Wang and Tchetgen Tchetgen [54], who introduce an alternative identification strategy based on an unconfounded compliance assumption. This allows point identification of the ATE while separating identification assumptions from estimation model assumptions. Their approach avoids reliance on parametric SEMs for ATE estimation and instead yields an efficient semiparametric estimator with an efficient influence function (EIF) that confers the estimator a multiple-robust property, *i.e.* the estimator remains consistent if one or several nuisance components are misspecified. This structure makes the estimators well-suited to nonparametric plug-in estimation using modern machine learning tools. This insight has been extended to develop multiply-robust CATE estimators that leverage binary instruments to adjust for unobserved confounding [18], and to debias confounded observational data by incorporating (potentially weak or imperfect) instruments [43]. Other recent work extends the framework of Wang and Tchetgen Tchetgen [54] to nonparametric identification of ATEs under related assumptions [34].

We adopt the framework of Wang and Tchetgen Tchetgen [54] as the foundation for our estimator, with the goal of efficiently and robustly estimating the ATE under adaptive, sequential data collection. Specifically, we derive the semiparametric efficiency bound for ATE estimation under arbitrary, covariate-dependent instrument-assignment policies and identify the optimal adaptive policy that minimizes this bound. We then introduce AMRIV, an adaptive, multiply robust estimator that attains the bound and enables valid inference in sequential experiments.

**Adaptive Experimental Design for Treatment Effect Estimation.** A substantial literature studies adaptive algorithms for estimating the average treatment effect (ATE) efficiently and with minimal variance when the treatment itself can be directly assigned. This line of work was initiated by Hahn et al. [23], who proposed a two-stage explore-then-commit design that asymptotically achieves the semiparametric efficiency bound, echoing early bandit-style adaptive allocation schemes. Fully sequential designs soon followed: Kato et al. [27] introduced the Adaptive Augmented Inverse Propensity Weighting (A2IPW) estimator, which achieves variance-optimal Neyman allocation; Kato et al. [28] extended this to settings with estimated policies and nuisance functions and demonstrated multiply robust consistency; and Cook et al. [13] added principled policy truncation and developed the first anytime-valid confidence sequences for adaptive ATE estimation.

Parallel progress has come from an online-learning perspective. Recent methods such as Clip-OGD and its optimistic variants attain sublinear or logarithmic "Neyman regret" for the ATE [14, 38, 39]; low-switching policies achieve finite-sample optimality bounds [35]; and newer designs jointly optimize over covariate and treatment dimensions [29]. Off-policy estimators with adaptively collected data have also obtained sharp error rates and regret guarantees [33]. Together, these advances form a mature toolkit for adaptive experimentation that combines adaptive nuisance learning, cross-fitting, policy truncation, regret-minimizing allocation, and time-uniform inference.

Our work builds directly on this literature but extends it to the far less explored setting where *only an instrument can be assigned* and treatment uptake is endogenous. We integrate core components from the direct-treatment literature—influence-function–based estimation, adaptive policy learning, cross-fitting, policy truncation, and sequential inference—to construct a unified estimator that retains semiparametric efficiency, multiply robust consistency, and time-uniform inference under noncompliance. Specifically, we are the first to: (i) derive the semiparametric efficiency bound for ATE estimation when only an instrument can be adaptively assigned; (ii) identify the variance-optimal allocation rule that balances outcome noise and compliance variability; and (iii) develop AMRIV, a multiply robust estimator that attains this bound and supports anytime-valid inference.

**Adaptive Experimentation with Instrumental Variables.** Recent work has begun to explore adaptive experimentation in settings where treatments cannot be directly assigned, requiring the use of instrumental variables (IVs) to estimate causal effects under unobserved confounding. Broadly, these efforts fall into two categories: methods aimed at improving predictive accuracy in the presence of confounding, and approaches focused on adaptive design and data collection or regret minimization using bandit-style feedback.

The first group focuses on improving estimation efficiency in indirect experiments. Gupta et al. [22] propose an adaptive framework for selecting among multiple data sources to efficiently estimate causal functionals such as the ATE. Their method, Online Moment Selection (OMS), chooses which source to query at each step based on moment conditions implied by a causal graph. While they address efficient data acquisition under structural constraints, their setting assumes passive data collection and differs from our focus on adaptive experimental design with noncompliance and endogenous treatment. Ailer et al. [3] study sequential indirect experiment design in instrumental variable settings, focusing on partial identification of nonlinear treatment effect queries. Rather than aiming for point estimation, their method adaptively tightens upper and lower bounds on a functional $Q[f]$ of the treatment effect by selecting experiments that reduce the gap between these bounds. In contrast, our work targets point identification and estimation of the ATE, and provides semiparametric efficiency and robustness guarantees under adaptive instrument assignment. Most closely related to our setting, Chandak et al. [9] study adaptive instrument selection to improve sample efficiency in indirect experiments. They propose a general influence-function–based optimization procedure for selecting instruments that minimize the mean squared error of nonparametric IV estimators, such as DeepIV [24]. However, their objective is variance reduction for prediction, not inference for causal estimands like the ATE. Their analysis is estimator-specific and does not characterize semiparametric efficiency bounds or multiply robust inference, which are central to our approach.

The second line of work focuses on regret minimization in settings with instrumental feedback. Zhao et al. [61] use randomized instruments within a linear structural equation model to enable pure exploration for policy learning under unobserved confounding. Their focus is on identifying the best treatment arm using bandit-style algorithms with finite-sample confidence intervals and near-optimal sample complexity guarantees. Unlike our work, which targets semiparametric inference for the ATE, their goal is policy optimization rather than estimation. Della Vecchia and Basu [16] study online instrumental variable regression with bandit feedback and propose regret bounds under endogeneity. Their focus is on prediction in stochastic settings with instrumental bandit structure, rather than causal effect estimation or statistical inference.

**Our Contribution.** To our knowledge, we present the first estimator that is both semiparametrically efficient and multiply robust for ATE estimation under a binary instrument with adaptive assignment. We characterize the efficiency bound, derive the optimal allocation policy that balances outcome and compliance variance, and develop the AMRIV estimator that asymptotically attains this bound. Our results generalize prior work on ATE estimation to settings with endogenous treatments, establish asymptotic normality, and construct time-uniform asymptotic confidence sequences. Empirical studies confirm our method's efficiency, robustness, and practical viability.

## A.2 Auxiliary Context

**Semiparametric Efficiency and Influence-Function–Based Methods.**

Our estimator builds on a long line of semiparametric inference techniques, particularly those using influence functions to achieve efficiency and robustness in the presence of nuisance components [7, 47]. Recent work has adapted these methods to flexible machine learning settings by incorporating sample-splitting and cross-fitting [11, 18, 30]. In adaptive experiments, such techniques have been

shown to yield efficient estimators without requiring Donsker conditions [13, 27, 28]. We extend these tools to a setting with endogenous treatment and adaptive assignment via a binary instrument.

### Multiply Robust Estimation.

Multiply robust estimators remain consistent if any one of multiple nuisance components is correctly specified. In the IV context, this structure has been exploited to enhance robustness of ATE and CATE estimators [18, 54]. We extend these ideas to adaptive settings, showing that AMRIV retains consistency even when some nuisance functions are misspecified, as long as at least one of $\delta(X)$ or $\delta^A(X)$ is consistently estimated.

### Confidence Sequences and Anytime-Valid Inference.

Confidence sequences (CSs) provide coverage guarantees that hold uniformly over time, making them well-suited to adaptive experiments with interim monitoring or early stopping. Recent work has developed CSs for influence-function–based estimators using martingale techniques and empirical Bernstein bounds [13, 25, 56]. We build on this to construct asymptotically valid confidence sequences for our adaptive IV estimator, accounting for sequential dependence and cross-fitted nuisances.

## B   Notation

Table 1: Notation

| | |
|---|---|
| $X$ | Observed covariates (feature vector) in $\mathbb{R}^m$. |
| $Z$ | Binary instrument / encouragement, $Z \in \{0, 1\}$ (experimenter-assigned). |
| $A$ | Binary treatment, $A \in \{0, 1\}$, determined endogenously by the unit. |
| $Y$ | Real-valued observed outcome. |
| $Y(a), Y(a, z)$ | Potential outcome under treatment $a$ (and instrument $z$ when shown). |
| $A(z)$ | Potential treatment taken under instrument $z$. |
| $U$ | Unobserved confounder(s). |
| $P$ | True (observable) distribution of $(X, Z, A, Y)$. |
| $\tau$ | Population average treatment effect (ATE): $\tau := \mathbb{E}[Y(1)] - \mathbb{E}[Y(0)]$. |
| $\mu^Y(z, x)$ | $\mathbb{E}[Y \mid Z = z, X = x]$, instrument-conditional outcome regression. |
| $\mu^A(z, x)$ | $\mathbb{E}[A \mid Z = z, X = x]$, instrument-conditional treatment regression. |
| $\delta^Y(x), \delta^A(x)$ | Instrument-induced shifts in $Y$ and $A$, respectively. |
| | $\delta^Y(x) = \mu^Y(1, x) - \mu^Y(0, x), \delta^A(x) = \mu^A(1, x) - \mu^A(0, x)$: |
| $\delta(x)$ | Conditional average treatment effect (CATE) at $x$: $\delta(x) := \delta^Y(x)/\delta^A(x)$. |
| $\pi_t(x \mid \mathcal{H}_{t-1})$ | Instrument assignment policy at time $t$: $\Pr(Z_t = 1 \mid X_t = x, \mathcal{H}_{t-1})$. |
| $\pi^*(x)$ | Efficiency-optimal instrument assignment (Eq. (4)). |
| $\mathcal{H}_{t-1}$ | History before round $t$: $\{(X_i, Z_i, A_i, Y_i)\}_{i=1}^{t-1}$. |
| $T$ | Total number of rounds / samples. |
| $T_0$ | Burn-in rounds (initial exploration). |
| $k_t$ | Truncation parameter at time $t$ (ensures $\pi_t \in [1/k_t, 1 - 1/k_t]$). |
| $\sigma^2(z, x)$ | Residual variance: $\text{Var}(Y - A\delta(X) \mid Z = z, X = x)$. |
| $V_{\text{eff}}(\pi)$ | Semiparametric efficiency bound under policy $\pi$ (Eq. (3)). |
| $\phi(\cdot; \pi, \eta)$ | Recentered efficient influence function (EIF) used in AMRIV (Eq. (2)). |
| $\widehat{f}, \widetilde{f}$ | $\widehat{f}$: estimate of $f$; $\widetilde{f}$: plug-in or candidate function. |
| $\widehat{\tau}_T^{\text{AMRIV}}$ | AMRIV estimator after $T$ rounds (Alg. 1). |
| $\widehat{\mathbb{E}}$ | Empirical expectation (sample average). |
| $\|g\|_2$ | $L_2$ norm: $\|g\|_2 := \mathbb{E}[g(X)^2]^{1/2}$. |
| $o_p(1), O_p(\cdot)$ | Standard probabilistic asymptotic order notation. |
| $\varepsilon$ | Small positive constant (e.g., variance-floor in Eq. (6)). |
| $\mathcal{H}_{t-1}^{(j)}$ | Temporal cross-fitting fold (e.g., $j \in \{0, 1\}$) used for sequential cross-fitting. |

## C   Practical Implementation of Nuisance and Variance Estimators

This section complements Remark 2 by outlining practical choices for nuisance and variance estimators, including nonnegativity and online-update considerations.

**Nuisance estimators.** Because AMRIV uses sequential cross-fitting, we require only standard nonparametric convergence rates to establish asymptotic normality (Theorems 3–4). Consequently, the analyst may flexibly choose estimators according to data structure and sample size. Representative options include:

Table 2: Representative nuisance estimators, convergence rates, and suitable applications.

| Estimator | Convergence rate | Best suited for |
|---|---|---|
| $k$-NN / kernel smoother | $O\!\left(n^{-\beta/(2\beta+d)}\right)$ for Hölder-$\beta$ smooth functions [41] | Low-dimensional, smooth problems |
| Random forest | $\tilde{O}\!\left(n^{-\beta/(2\beta+d)}\right)$ for Hölder-$\beta$ smooth functions [49, 53] | Moderate-dimensional, tabular data |
| Fully connected neural nets (ReLU) | $O\!\left(\sqrt{WL\log W}\, n^{-1/2}\right)$ for width $W$ and depth $L$ [59] | High-dimensional or structured inputs |
| Neural nets (1-Lipschitz, bounded weights) | $O\!\left(\sqrt{\prod_{l=1}^{L} M_l}/n^{1/4}\right)$ where $M_l$ bounds the Frobenius norm of layer $l$'s weights [20] | High-dimensional or structured inputs |

The best choice depends on sample size, covariate dimension, and smoothness. In adaptive experiments with directly assigned treatments, $k$-NN and random forests are standard baselines [13, 27].

**Online or streaming implementations.** Theoretical results assume that nuisance functions are re-estimated at each update, but full data storage is not required. Practical alternatives include:

1. **Online learners:** gradient-descent (SGD) updates, online random forests [32], or GLMs can update parameters incrementally using new data.
2. **Block sample-splitting:** reuse recent batches to update nuisance fits and discard older data while maintaining cross-fitting validity [26].
3. **Sufficient-statistic updates:** for parametric or binned regressors, sufficient statistics such as running sums of $(X, Z, A, Y)$ per cell suffice.

**Non-negative variance estimation.** Equation (6) enforces non-negativity via a small floor parameter $\varepsilon$. As an alternative, one may use a *self-normalized kernel variance estimator* that is fully nonparametric and guarantees $\widehat{\sigma}_t^2(z, x) \geq 0$ by construction.

For each $(z, x)$, define

$$\widehat{\sigma}_t^2(z, x) = \frac{\sum_{s \leq t} K_h(\|X_s - x\|)\,\mathbb{I}\{Z_s = z\}\,(R_s - \widehat{\mu}_{t-1}(z, x))^2}{\sum_{s \leq t} K_h(\|X_s - x\|)\,\mathbb{I}\{Z_s = z\}},$$

$$\widehat{\mu}_{t-1}(z, x) = \frac{\sum_{s \leq t} K_h(\|X_s - x\|)\,\mathbb{I}\{Z_s = z\}\,R_s}{\sum_{s \leq t} K_h(\|X_s - x\|)\,\mathbb{I}\{Z_s = z\}},$$

where $R_s = Y_s - A_s \widehat{\delta}_{t-1}(X_s)$ and $K_h(\cdot)$ is a kernel with bandwidth $h$.

This estimator satisfies $\widehat{\sigma}_t^2(z, x) \geq 0$ by construction and is consistent for $\mathrm{Var}(Y - A\delta(X) \mid Z = z, X = x)$ under standard kernel conditions. **Caveat:** updating kernel weights online can be $O(t^2)$ in general; for large-scale data, the plug-in estimator in Eq. (6) is more computationally efficient.

Both the plug-in estimator with the floor $\varepsilon$ and the kernel-based version above preserve all asymptotic guarantees; the choice between them is primarily a trade-off between computational efficiency and strict non-negativity.

# D   Asymptotic Confidence Sequences

The fixed–time intervals in Section 6 guarantee $(1 - \alpha)$ coverage *solely* at a pre-specified sample size $T$. In practice, however, analysts often *peek* at interim results and may stop the study early once

a decision rule is met [46], behavior that invalidates fixed-time intervals. To remain valid under such data-dependent stopping one needs a **confidence sequence (CS)**—a collection of intervals

$$[L_t,\ U_t]_{t\geq 1}$$

satisfying the time-uniform guarantee

$$P\left(\forall t \in \mathbb{N}^+ : \tau \in [L_t, U_t]\right)\ \geq\ 1 - \alpha.$$

Constructing *non-asymptotic*, anytime-valid CSs can be difficult when the target estimand contains estimated nuisance functions. Fortunately, for AMRIV the nuisance-induced remainder is $o_p(t^{-1/2})$ under Theorem 3 assumptions, so an *asymptotic* CS—valid after a finite, burn-in phase— remains both tractable and practically useful.

**Definition 1** (Asymptotic time-uniform coverage [15, Def. 2.1 & 2.3])**.** *A sequence of random intervals $\widetilde{C}_t = [\widetilde{L}_t, \widetilde{U}_t]_{t\geq 1}$ is an asymptotic time uniform $(1-\alpha)$ confidence sequence (AsympCS) for a parameter $\tau$ if the following two conditions hold.*

*(i) **Asymptotic confidence sequence:** there exists an exact (potentially unknown) $(1-\alpha)$ confidence sequence $C_t^\star = [L_t^\star, U_t^\star]_{t\geq 1}$ such that $\widetilde{L}_t/L_t^\star \to 1$ and $\widetilde{U}_t/U_t^\star \to 1$ almost surely.*

*(ii) **Asymptotic time-uniform coverage:***

$$\lim_{T_0 \to \infty} P\left(\forall t \geq T_0 : \tau \in \widetilde{C}_t\right)\ \geq\ 1 - \alpha.$$

Definition 1 can be read as follows: if one waits to "peek" until the sample size is sufficiently large ($T \geq T_0$ for some burn-in $T_0$), the band then covers the true parameter at *every* later time with probability approaching $1 - \alpha$. In practice, the rare coverage failures occur almost exclusively during this short initial window; once past it, the intervals tighten rapidly and deliver appreciable power gains over fully non-asymptotic sequences [13, 55].

Building on the methodologies of Waudby-Smith et al. [56] and Cook et al. [13], we now present the corresponding asymptotic confidence-sequence (AsympCS) results for our estimator:

**Theorem 6** (AsympCS for AMRIV)**.** *Suppose Assumptions 1 to 3 hold and there exists a non-adaptive policy $\pi(X) \in [\epsilon, 1-\epsilon]$ for some $\epsilon > 0$ such that the nuisances estimates $\widehat{\eta}_t$ and the adaptive assignment policy $\pi_t(X \mid \mathcal{H}_{t-1})$ are $L_2$-consistent relative to the truncation schedule, i.e. $k_t\|\widehat{f}_{t-1} - f\|_2 = o_p(1)$ and $k_t\|\pi_t - \pi\|_2 = o_p(1)$ for $f \in \{\mu^Y(0,\cdot), \mu^A(0,\cdot), \delta(\cdot), \delta^A(\cdot)\}$. Furthermore, assume $\|\hat{\delta}_{t-1} - \delta\|_2 = o_{a.s.}(1)$ and $\|\hat{\delta}_{t-1} - \delta\|_2\|\hat{\delta}_{t-1}^A - \delta^A\|_2 = o_{a.s.}\left(\sqrt{\frac{\log t}{t}}\right)$. Let*

$$\widehat{V}_T := \frac{1}{T}\sum_{t=1}^{T}\left(\phi(X_t, Z_t, A_t, Y_t;\ \pi_t, \widehat{\eta}_t) - \widehat{\tau}_T^{AMRIV}\right)^2,$$

*be the estimated variance of $\{\phi(X_t, Z_t, A_t, Y_t;\ \pi_t, \widehat{\eta}_t)\}$, and fix a user-specified $\rho > 0$. Then, for all $T > 0$, the interval*

$$\tilde{C}_T^{\mathrm{AsympCS}} := \left(\widehat{\tau}_T^{AMRIV}\ \pm\ \sqrt{\frac{2(T\widehat{V}_T\rho^2 + 1)}{T^2\rho^2}\ \log\left(\frac{\sqrt{T\widehat{V}_T\rho^2 + 1}}{\alpha}\right)}\right)$$

*forms an asymptotic $(1-\alpha)$ confidence sequence (as in Definition 1) for $\tau$. Furthermore, width of $\tilde{C}_T^{\mathrm{AsympCS}}$ is (approximately) minimized at*

$$\rho^\star\ =\ \sqrt{\frac{-2\log\alpha\ +\ \log(-2\log\alpha + 1)}{T}}.$$

**Remark 3** (Difference in convergence rates for fixed-time and anytime-valid inference.)**.** *The conditions under which the AsympCS in Theorem 6 is valid are largely the same as those imposed for fixed time inference in Theorem 3. The main difference lies in the convergence conditions for the nuisance functions. While $\|\hat{\delta}_{t-1} - \delta\|_2$ and $\|\hat{\delta}_{t-1} - \delta\|_2\|\hat{\delta}_{t-1}^A - \delta^A\|_2$ are assumed to converge in probability for fixed time inference, confidence sequences require convergence almost surely, as noted by Waudby-Smith et al. [55]. However, the conditions for valid inference are not necessarily stricter than fixed-time inference, as the product error term is allowed to converge a slower $\sqrt{\log t/t}$ rate as opposed to $t^{-1/2}$.*

*Proof of Theorem 6.* For the proof of Theorem 6, we rely on an existing result for asymptotic confidence sequences [55]. For completeness, we provide this result in Lemma 7 below.

**Lemma 7** (Corollary 3.4 of Waudby-Smith et al. [55] )**.** *Suppose $\hat{\theta}_t$ is an asymptotically linear estimator of $\theta$ with influence function $\phi$ that satisfies*

$$\hat{\theta}_t - \theta = \frac{1}{t} \sum_{i=1}^{t} \phi(X_i, Z_i, A_i, Y_i; \pi_i, \eta) + o_{a.s.} \left( \sqrt{\log t/t} \right). \tag{12}$$

*Furthermore, suppose that $Var(\phi) < \infty$. Then, $\left( \hat{\theta}_t \pm \sqrt{\frac{2(t\rho^2+1)}{t^2\rho^2} \log \left( \frac{\sqrt{t\rho^2+1}}{\alpha} \right)} \right)$ forms a valid*

*$(1 - \alpha)$-AsympCS (as in Definition 1) for $\theta$.*

Using Lemma 7, we only need to show that (i) the residual error of our estimator is of a smaller order than $\sqrt{\log t/t}$ almost surely and (ii) the variance of the limiting influence function $\phi$ is bounded.

**Verifying Residual Error.** Our proof of the residual error bound follows similar steps to the proof of Theorem 3. We first rewrite the difference between our estimate $\hat{\tau}_t^{AMRIV}$ and $\tau$ as

$$\hat{\tau}_t^{AMRIV} - \tau = \frac{1}{t} \sum_{i=1}^{t} \phi(X_i, A_i, Z_i, Y_i; \pi_i, \eta) - \frac{1}{t} \sum_{i=1}^{t} m_t, \tag{13}$$

where $m_t = \phi(X_i, A_i, Z_i, Y_i; \pi_i, \hat{\eta}_i) - \phi(X_i, A_i, Z_i, Y_i; \pi_t, \eta)$. We repeat the same arguments as the proof of Theorem 3, which shows that the cumulative residual error (i.e. the sum of $m_t$) vanish at $o_p(1/\sqrt{t})$ rates. Replacing assumptions $\|\hat{\delta}_{t-1} - \delta\|_2 = o_p(1)$ with $\|\hat{\delta}_{t-1} - \delta\|_2 = o_{a.s.}(1)$ and $\|\hat{\delta}_{t-1} - \delta\|_2 \|\hat{\delta}_{t-1}^A - \delta^A\|_2 = o_p(t^{-1/2})$ with $\|\hat{\delta}_{t-1} - \delta\|_2 \|\hat{\delta}_{t-1}^A - \delta^A\|_2 = o_{a.s.} \left( \sqrt{\frac{\log t}{t}} \right)$, we obtain $\sum_{i=1}^{t} m_t = o_{a.s.}(\sqrt{t \log t})$. Normalizing by $t$, we obtain the desired result.

**Finite Variance of $\phi$.** The finite variance of limiting influence function $\phi$ is immediate from Assumption 3 and the condition that $\pi(X) \in [\epsilon, 1 - \epsilon]$ for some $\epsilon > 0$. Under these assumptions, for any tuple $(X, Z, A, Y)$, $\phi$ is bounded almost surely as a function of some constant $C$,

$$|\phi(X, Z, A, Y; \pi, \eta)| = \frac{2}{\epsilon} \left( 3C^2 + C^3 \right) + C.$$

Using this bound, we now upper bound the variance as follows:

$$\text{Var} (\phi) = \mathbb{E}[\phi^2] - \mathbb{E}[\phi]^2 \le \mathbb{E}[\phi^2] \le \left( \frac{2}{\epsilon} \left( 3C^2 + C^3 \right) + C \right)^2.$$

Because the constant $C$ is finite, the variance must also be finite, which completes our proof. By Lemma 7, the confidence sequence in Theorem 6 is a valid $(1 - \alpha)$-AsympCS for $\tau$. The proof for the approximate choice of $\rho^*$ that minimizes the relative width of the confidence sequence at time $T$ is provided in Waudby-Smith et al. [55, Appendix B.2]. □

# E   Proof of Theorem 1 and Corollary 2

In semiparametric theory, the efficiency bound is determined by the variance of the EIF characterized in Eq. (2):

$$\begin{aligned} V_{\text{eff}}(\pi) &= \mathbb{E}[(\phi(X, Z, A, Y; \eta) - \tau)^2] \\ &= \mathbb{E} \left[ \mathbb{E}[(\phi(X, Z, A, Y; \eta) - \tau)^2 \mid X] \right] \quad \text{(Law of iterated expectations)} \end{aligned}$$

where

$$\begin{aligned} &\phi(X, Z, A, Y; \eta) - \tau \\ &= \underbrace{\frac{2Z - 1}{Z\pi(X) + (1 - Z)(1 - \pi(X))} \frac{1}{\delta^A(X)} \left[ Y - A\delta(X) - \mu^Y(0, X) + \mu^A(0, X)\delta(X) \right] + \delta(X) - \tau}_{\Lambda_1} \end{aligned}$$

First, we show that $\mathbb{E}[\Lambda_1 \mid X] = 0$:

$$\mathbb{E}[\Lambda_1 \mid X] = \pi(X)\mathbb{E}[\Lambda_1 \mid Z = 1, X] + (1 - \pi(X))\mathbb{E}[\Lambda_1 \mid Z = 0, X]$$
$$= \frac{\pi(X)}{\pi(X)}\frac{1}{\delta^A(X)}\left(\mu^Y(1, X) - \mu^A(1, X)\delta(X) - \mu^Y(0, X) + \mu^A(0, X)\delta(X)\right)$$
$$+ \frac{1 - \pi(X)}{1 - \pi(X)}\frac{1}{\delta^A(X)}\underbrace{\left(\mu^Y(0, X) - \mu^A(0, X)\delta(X) - \mu^Y(0, X) + \mu^A(0, X)\delta(X)\right)}_{=0}$$

(Using the $\mu^A, \mu^Y$ definitions)

$$= \frac{1}{\delta^A(X)}(\mu^Y(1, X) - \mu^A(1, X)\delta(X) - \mu^Y(0, X) + \mu^A(0, X)\delta(X))$$
$$= 0$$

where in the last line we used the fact that $\delta(X) = \frac{\mu^Y(1,X) - \mu^Y(0,X)}{\mu^A(1,X) - \mu^A(0,X)}$ which implies the identity
$\mu^Y(1, X) - \mu^A(1, X)\delta(X) = \mu^Y(0, X) - \mu^A(0, X)\delta(X)$.

Thus, we can expand $V_{\text{eff}}(\pi)$ as:

$$V_{\text{eff}}(\pi) = \mathbb{E}\left[\mathbb{E}[(\Lambda_1 + \delta(X) - \tau)^2 \mid X]\right]$$
$$= \mathbb{E}\left[\mathbb{E}[\Lambda_1^2 \mid X] + (\delta(X) - \tau)^2\right] - 2\mathbb{E}\left[(\delta(X) - \tau)\mathbb{E}[\Lambda_1 \mid X]\right]$$
$$= \mathbb{E}\left[\mathbb{E}[\Lambda_1^2 \mid X] + (\delta(X) - \tau)^2\right] \qquad \text{(Using } \mathbb{E}[\Lambda_1 \mid X]\text{=0)}$$

It now remains to expand $\mathbb{E}[\Lambda_1^2 \mid X]$:

$$\mathbb{E}[\Lambda_1^2 \mid X] = \pi(X)\mathbb{E}[\Lambda_1^2 \mid Z = 1, X] + (1 - \pi(X))\mathbb{E}[\Lambda_1^2 \mid Z = 0, X]$$
$$= \frac{\pi(X)}{\pi(X)^2}\frac{1}{\delta^A(X)^2}\mathbb{E}\left[(Y - A\delta(X) - \mu^Y(0, X) + \mu^A(0, X)\delta(X))^2 \mid Z = 1, X\right]$$
$$+ \frac{1 - \pi(X)}{(1 - \pi(X))^2}\frac{1}{\delta^A(X)^2}\mathbb{E}\left[(Y - A\delta(X) - \mu^Y(0, X) + \mu^A(0, X)\delta(X))^2 \mid Z = 0, X\right]$$
$$= \frac{1}{\delta^A(X)^2}\left(\frac{1}{\pi(X)}\text{Var}(Y - A\delta(X) \mid Z = 1, X)\right.$$
$$\left. + \frac{1}{1 - \pi(X)}\text{Var}(Y - A\delta(X) \mid Z = 0, X)\right)$$
$$= \frac{1}{\delta^A(X)^2}\left(\frac{\sigma^2(1, X)}{\pi(X)} + \frac{\sigma^2(0, X)}{1 - \pi(X)}\right)$$

where we used the fact that $\mathbb{E}[Y - A\delta(X) \mid Z = 0, X] = \mu^Y(0, X) - \mu^A(0, X)\delta(X)$ and
$\mathbb{E}[Y - A\delta(X) \mid Z = 1, X] = \mu^Y(1, X) - \mu^A(1, X)\delta(X) = \mu^Y(0, X) - \mu^A(0, X)\delta(X)$. Putting
everything together, we obtain the result of Theorem 1:

$$V_{\text{eff}}(\pi) := \mathbb{E}\left[\frac{1}{\delta^A(X)^2}\left(\frac{\sigma^2(1, X)}{\pi(X)} + \frac{\sigma^2(0, X)}{1 - \pi(X)}\right) + (\delta(X) - \tau)^2\right].$$

Then, the optimal policy $\pi^*(X)$ is given by:

$$\pi^* = \arg\min_\pi V_{\text{eff}}(\pi)$$
$$= \arg\min_\pi \mathbb{E}\left[\frac{1}{\delta^A(X)^2}\left(\frac{\sigma^2(1, X)}{\pi(X)} + \frac{\sigma^2(0, X)}{1 - \pi(X)}\right)\right]$$
$$= \arg\min_\pi \mathbb{E}\left[\frac{1}{\delta^A(X)^2}\mathbb{E}\left[\left(\frac{\sigma^2(1, X)}{\pi(X)} + \frac{\sigma^2(0, X)}{1 - \pi(X)}\right)\Big| X\right]\right]$$
$$\Rightarrow \pi^*(X) = \arg\min_p \left(\frac{\sigma^2(1, X)}{p} + \frac{\sigma^2(0, X)}{1 - p}\right)$$

The minimum is obtained when the derivative of the argument w.r.t. $p$ is 0, *i.e.* $\frac{\sigma^2(0,X)}{(1-p)^2} - \frac{\sigma^2(1,X)}{p^2} = 0$.

By solving for $p$, we obtain $p = \frac{\sqrt{\sigma^2(1,X)}}{\sqrt{\sigma^2(1,X)} + \sqrt{\sigma^2(0,X)}}$. Thus, we obtain the result of Corollary 2:

$$\pi^*(X) = \frac{\sqrt{\sigma^2(1,X)}}{\sqrt{\sigma^2(1,X)} + \sqrt{\sigma^2(0,X)}}.$$

# F   Proof of Theorem 3

## F.1   Preliminaries

Our asymptotic argument relies on a martingale central limit theorem under a Lindeberg-type condition. We use a streamlined version of the MDS central limit theorem originally due to Dvoretzky [17], as presented in Zhang et al. [60, Theorem 2].

**Theorem 8** (Martingale CLT, adapted from [60, Thm. 2])**.** *Let $\{(z_t, \mathcal{H}_t)\}_{t=1}^T$ be a real-valued sequence where $\bar{z}_T = \frac{1}{T}\sum_{t=1}^T z_t$ such that:*

1. *(Martingale difference sequence) $\{z_t\}_{t=1}^T$ is a martingale difference sequence; that is, $\mathbb{E}[z_t \mid \mathcal{H}_{t-1}] = 0$ for every $t \in [1,T]$.*

2. *(Conditional variance convergence) There exists a constant $\sigma^2 > 0$ such that*

$$\frac{1}{T}\sum_{t=1}^T \mathbb{E}[z_t^2 \mid \mathcal{H}_{t-1}] \xrightarrow{p} \sigma^2,$$

3. *(Lindeberg condition) For every $\epsilon > 0$,*

$$\frac{1}{T}\sum_{t=1}^T \mathbb{E}\left[z_t^2 \mathbb{I}\{|z_t| > \epsilon\sqrt{T}\} \mid \mathcal{H}_{t-1}\right] \xrightarrow{p} 0.$$

*Then,*

$$\sqrt{T}\,\bar{z}_T \xrightarrow{d} \mathcal{N}(0,\sigma^2).$$

To begin our proof, we define $\psi_t := \phi(X_t, Z_t, A_t, Y_t; \pi_t, \widehat{\eta}_t) - \tau$, where $\phi$ is given in Eq. (9). Letting $\eta = \{\mu^Y(0,X), \mu^A(0,X), \delta^A(X), \delta(X)\}$ denote the true nuisance values, we decompose $\psi_t$ as

$$\psi_t = \underbrace{\phi(X_t, Z_t, A_t, Y_t; \pi_t, \eta) - \tau}_{z_t} + \underbrace{\phi(X_t, Z_t, A_t, Y_t; \pi_t, \widehat{\eta}_t) - \phi(X_t, Z_t, A_t, Y_t; \pi_t, \eta)}_{m_t},$$

such that

$$\sqrt{T}(\widehat{\tau}_T^{\text{AMRIV}} - \tau) = \sqrt{T}\left(\frac{1}{T}\sum_{t=1}^T z_t\right) + \sqrt{T}\left(\frac{1}{T}\sum_{t=1}^T m_t\right).$$

Then, the proof of Theorem 3 proceeds in three steps. We first verify that $\{z_t\}_{t=1}^T$ forms a martingale difference sequence. Then, we show that $\{z_t\}_{t=1}^T$ satisfies conditions (2)-(3) of Theorem 8 with $\sigma^2 = V_{\text{eff}}(\pi)$. Finally, we will show that $\sqrt{T}(\frac{1}{T}\sum_{t=1}^T m_t) = o_p(1)$, thus concluding that

$$\sqrt{T}(\widehat{\tau}_T^{\text{AMRIV}} - \tau) \xrightarrow{d} \mathcal{N}(0, V_{\text{eff}}(\pi)).$$

## F.2   MDS structure of $z_t$

We now show that $\{z_t\} = \{\phi_t(X_t, Z_t, A_t, Y_t; \pi_t, \eta) - \tau\}$ forms an MDS, *i.e.* $\mathbb{E}[z_t \mid \mathcal{H}_{t-1}] = 0$:

$$\mathbb{E}[z_t \mid \mathcal{H}_{t-1}]$$
$$= \mathbb{E}\Big[\frac{2Z_t - 1}{Z_t \pi_t(X_t \mid \mathcal{H}_{t-1}) + (1 - Z_t)(1 - \pi_t(X_t \mid \mathcal{H}_{t-1}))}\frac{1}{\delta^A(X_t)}$$

$$\cdot \left[ Y_t - A_t \delta(X_t) - \mu^Y(0, X_t) + \mu^A(0, X_t)\delta(X_t) \right] + \delta(X_t) \Big| \mathcal{H}_{t-1} \right] - \tau$$

$$= \mathbb{E}\left[ \mathbb{E}\left[ \frac{2Z_t - 1}{Z_t \pi_t(X_t \mid \mathcal{H}_{t-1}) + (1 - Z_t)(1 - \pi_t(X_t \mid \mathcal{H}_{t-1}))} \frac{1}{\delta^A(X_t)} \right.\right.$$

$$\left.\left. \cdot \left[ Y_t - A_t \delta(X_t) - \mu^Y(0, X_t) + \mu^A(0, X_t)\delta(X_t) \right] \Big| X_t, \mathcal{H}_{t-1} \right] \Big| \mathcal{H}_{t-1} \right] + \tau - \tau$$

From Eq. (1)

$$= \mathbb{E}\left[ \frac{1}{\delta^A(X_t)} \left[ \mu^Y(1, X_t) - \mu^A(1, X_t)\delta(X_t) - \mu^Y(0, X_t) + \mu^A(0, X_t)\delta(X_t) \right] \right.$$

$$\left. - \frac{1}{\delta^A(X_t)} \left[ \mu^Y(0, X_t) - \mu^A(0, X_t)\delta(X_t) - \mu^Y(0, X_t) + \mu^A(0, X_t)\delta(X_t) \right] \Big| \mathcal{H}_{t-1} \right]$$

$$= 0$$

where we used the identity $\mu^Y(1, X_t) - \mu^A(1, X_t)\delta(X_t) = \mu^Y(0, X_t) - \mu^A(0, X_t)\delta(X_t)$ which follows from the definition of $\delta(X_t)$. Thus, $\{z_t\}$ is an MDS, owing to the fact that $\pi_t$ is constructed from historical data only.

### F.3 $z_t$ satisfies conditions (2)–(3) of Theorem 8

For condition (2), we first show that $\mathbb{E}[z_t^2 \mid \mathcal{H}_{t-1}] - V_{\text{eff}}(\pi) \xrightarrow{P} 0$:

$$\mathbb{E}[z_t^2 \mid \mathcal{H}_{t-1}] - V_{\text{eff}}(\pi)$$

$$= \text{Var}(z_t \mid \mathcal{H}_{t-1}) - \mathbb{E}\left[ \frac{1}{\delta^A(X_t)^2} \left( \frac{\sigma^2(1, X_t)}{\pi(X_t)} + \frac{\sigma^2(0, X_t)}{1 - \pi(X_t)} \right) + (\delta(X_t) - \tau)^2 \right]$$

$(\mathbb{E}[z_t \mid \mathcal{H}_{t-1}] = 0)$

$$= \text{Var}(\phi(X_t, Z_t, A_t, Y_t; \pi_t, \eta) \mid \mathcal{H}_{t-1})$$

$$- \mathbb{E}\left[ \frac{1}{\delta^A(X_t)^2} \left( \frac{\sigma^2(1, X_t)}{\pi(X_t)} + \frac{\sigma^2(0, X_t)}{1 - \pi(X_t)} \right) + (\delta(X_t) - \tau)^2 \right]$$

$$= \mathbb{E}\left[ \frac{1}{\delta^A(X_t)^2} \left( \frac{\sigma^2(1, X_t)}{\pi_t(X_t \mid \mathcal{H}_{t-1})} + \frac{\sigma^2(0, X_t)}{1 - \pi_t(X_t \mid \mathcal{H}_{t-1})} \right) + (\delta(X_t) - \tau)^2 \Big| \mathcal{H}_{t-1} \right]$$

($\eta$ is the oracle nuisance set)

$$- \mathbb{E}\left[ \frac{1}{\delta^A(X_t)^2} \left( \frac{\sigma^2(1, X_t)}{\pi(X_t)} + \frac{\sigma^2(0, X_t)}{1 - \pi(X_t)} \right) + (\delta(X_t) - \tau)^2 \right]$$

$$= \mathbb{E}\left[ \frac{\sigma^2(1, X_t)}{\delta^A(X_t)^2} \left( \frac{\pi(X_t) - \pi_t(X_t \mid \mathcal{H}_{t-1})}{\pi_t(X_t \mid \mathcal{H}_{t-1})\pi(X_t)} \right) \Big| \mathcal{H}_{t-1} \right]$$

$$+ \mathbb{E}\left[ \frac{\sigma^2(0, X_t)}{\delta^A(X_t)^2} \left( \frac{\pi_t(X_t \mid \mathcal{H}_{t-1}) - \pi(X_t)}{(1 - \pi_t(X_t \mid \mathcal{H}_{t-1}))(1 - \pi(X_t))} \right) \Big| \mathcal{H}_{t-1} \right]$$

$$\leq \frac{36 C^2 \epsilon_{\delta^A}^2 k_t}{\epsilon} |\mathbb{E}[\pi(X_t) - \pi_t(X_t \mid \mathcal{H}_{t-1})]| \lesssim k_t \|\pi_t - \pi\|_2 = o_p(1)$$

where in the last line we use the following boundedness conditions: (i) $|Y| \leq C$ from Assumption 3 and thus $|\delta(X)| \leq 2C$ and $\sigma^2(z, X_t) = \mathbb{E}[(Y - A\delta(X_t))^2 \mid Z = z, X_t] - \mathbb{E}[Y - A\delta(X_t) \mid Z = z, X_t]^2 \leq 18C^2$, (ii) $|\delta^A(X)|^{-1} \leq \epsilon_{\delta^A}$ for some $\epsilon_{\delta^A} > 0$ implicit in the (conditional) relevance in Assumption 1, (iii) $\pi(X_t), 1 - \pi(X_t) > \epsilon$ from Theorem 3 statement, (iv) $\pi(X_t), 1 - \pi(X_t) \geq 1/k_t$ by construction, and (v) the $L_1$ norm is bounded by the $L_2$ norm. Thus, setting $\sigma^2 := V_{\text{eff}}$, we have that each term converges in probability to $\sigma^2$, *i.e.* $\mathbb{E}[z_t^2 \mid \mathcal{H}_{t-1}] \xrightarrow{P} \sigma^2$, where $\sigma^2$ is finite by Assumption 1 and Assumption 3.

To complete condition (2), we now show that

$$\left| \frac{1}{T} \sum_{t=1}^{T} \mathbb{E}[z_t^2 \mid \mathcal{H}_{t-1}] - \sigma^2 \right| \xrightarrow{P} 0.$$

Let $a_t := \mathbb{E}[z_t^2 \mid \mathcal{H}_{t-1}]$ and $a := \sigma^2$. We have just established that $a_t \xrightarrow{P} a$, and under our boundedness assumptions, $\sup_t \mathbb{E}[a_t] < \infty$, so the sequence $\{a_t\}$ is uniformly integrable. By the $L^1$ convergence theorem (e.g., Loève [36]), this implies that $a_t \to a$ in $L^1$, i.e., $\mathbb{E}\left[|\mathbb{E}[z_t^2 \mid \mathcal{H}_{t-1}] - \sigma^2|\right] \to 0$.

Therefore, by Cesàro averaging and Markov's inequality, we obtain $\frac{1}{T}\sum_{t=1}^{T}\mathbb{E}[z_t^2 \mid \mathcal{H}_{t-1}] \xrightarrow{p} \sigma^2$, completing the verification of condition (2) in Theorem 8.

Now we verify that $z_t$ satisfies condition (3), the Lindeberg condition. We follow the same steps as in Cook et al. [13]. Let $b_t := z_t^2 \cdot \mathbb{I}\{|z_t| > \delta\sqrt{T}\}$. Then $b_t = z_t^2$ with probability $\Pr(|z_t| > \delta\sqrt{T})$, and $b_t = 0$ otherwise. By Chebyshev's inequality,

$$\Pr(|z_t| > \delta\sqrt{T}) \leq \frac{\operatorname{Var}(z_t)}{\delta^2 T}.$$

Since $\operatorname{Var}(z_t) = \mathbb{E}[z_t^2] < \infty$, it follows that $\lim_{T\to\infty} \frac{\operatorname{Var}(z_t)}{\delta^2 T} = 0$, which implies $b_t \xrightarrow{p} 0$ and hence $b_t \xrightarrow{d} 0$. Moreover, note that $|b_t| \leq z_t^2$ and $\mathbb{E}[z_t^2] < \infty$. By the dominated convergence theorem,

$$\lim_{T\to\infty}\mathbb{E}[b_t] = \mathbb{E}\left[\lim_{T\to\infty} b_t\right] = 0.$$

Therefore,

$$\frac{1}{T}\sum_{t=1}^{T}\mathbb{E}\left[z_t^2 \cdot \mathbb{I}\left\{|z_t| > \delta\sqrt{T}\right\} \mid \mathcal{H}_{t-1}\right] \xrightarrow{p} 0,$$

verifying the Lindeberg-type condition required for the martingale CLT.

**F.4** $\quad \sqrt{T}\left(\frac{1}{T}\sum_{t=1}^{T} m_t\right)$ **is** $o_p(1)$

We first decompose $\sqrt{T}\left(\frac{1}{T}\sum_{t=1}^{T} m_t\right)$ as:

$$\sqrt{T}\left(\frac{1}{T}\sum_{t=1}^{T} m_t\right)$$
$$= \sqrt{T}\left(\frac{1}{T}\sum_{t=1}^{T}(\phi(X_t, Z_t, A_t, Y_t; \pi_t, \widehat{\eta}_t) - \phi(X_t, Z_t, A_t, Y_t; \pi_t, \eta))\right)$$
$$= \sqrt{T}\left(\frac{1}{T}\sum_{t=1}^{T}(\mathbb{E}[\phi(X_t, Z_t, A_t, Y_t; \pi_t, \widehat{\eta}_t) \mid \mathcal{H}_{t-1}] - \mathbb{E}[\phi(X_t, Z_t, A_t, Y_t; \pi_t, \eta) \mid \mathcal{H}_{t-1}])\right) \quad (\Delta^A)$$
$$+ \sqrt{T}\Big(\frac{1}{T}\sum_{t=1}^{T}\Big\{(\phi(X_t, Z_t, A_t, Y_t; \pi_t, \widehat{\eta}_t) - \phi(X_t, Z_t, A_t, Y_t; \pi_t, \eta))$$
$$- \mathbb{E}[\phi(X_t, Z_t, A_t, Y_t; \pi_t, \widehat{\eta}_t) - \phi(X_t, Z_t, A_t, Y_t; \pi_t, \eta) \mid \mathcal{H}_{t-1}]\Big\}\Big) \quad (\Delta^B)$$

where $\Delta^A$ is an asymptotic bias term due to nuisance estimation and $\Delta^B$ is the empirical process term. We bound these independently. Let $\Delta_t^A = \mathbb{E}[\phi(X_t, Z_t, A_t, Y_t; \pi_t, \widehat{\eta}_t) \mid \mathcal{H}_{t-1}] - \mathbb{E}[\phi(X_t, Z_t, A_t, Y_t; \pi_t, \eta) \mid \mathcal{H}_{t-1}]$. Then:

$$\Delta_t^A$$
$$= \mathbb{E}[\phi(X_t, Z_t, A_t, Y_t; \pi_t, \widehat{\eta}_t) \mid \mathcal{H}_{t-1}] - \mathbb{E}[\phi(X_t, Z_t, A_t, Y_t; \pi_t, \eta) \mid \mathcal{H}_{t-1}]$$
$$= \mathbb{E}[\mathbb{E}[\phi(X_t, Z_t, A_t, Y_t; \pi_t, \widehat{\eta}_t) \mid X_t, \mathcal{H}_{t-1}] \mid \mathcal{H}_{t-1}] - \mathbb{E}[\delta(X_t) \mid \mathcal{H}_{t-1}]$$
$$= \mathbb{E}\big[\mathbb{E}[\phi(X_t, Z_t, A_t, Y_t; \pi_t, \widehat{\eta}_t) \mid Z_t = 1, X_t, \mathcal{H}_{t-1}]\pi_t(X_t \mid \mathcal{H}_{t-1}) \mid \mathcal{H}_{t-1}\big]$$
$$\quad + \mathbb{E}\big[\mathbb{E}[\phi(X_t, Z_t, A_t, Y_t; \pi_t, \widehat{\eta}_t) \mid Z_t = 0, X_t, \mathcal{H}_{t-1}](1 - \pi_t(X_t \mid \mathcal{H}_{t-1})) \mid \mathcal{H}_{t-1}\big]$$
$$\quad - \mathbb{E}[\delta(X_t) \mid \mathcal{H}_{t-1}]$$
$$= \mathbb{E}\Bigg[\frac{1}{\widehat{\delta}_{t-1}^A(X_t)}(\mu^Y(1, X_t) - \mu^A(1, X_t)\widehat{\delta}_{t-1}(X_t) + \widehat{\mu}_{t-1}^Y(0, X_t) - \widehat{\mu}_{t-1}^A(0, X_t)\widehat{\delta}_{t-1}(X_t))$$
$$\quad - \frac{1}{\widehat{\delta}_{t-1}^A(X_t)}(\mu^Y(0, X_t) - \mu^A(0, X_t)\widehat{\delta}_{t-1}(X_t) + \widehat{\mu}_{t-1}^Y(0, X_t) - \widehat{\mu}_{t-1}^A(0, X_t)\widehat{\delta}_{t-1}(X_t))$$

$$+ \widehat{\delta}_{t-1}(X_t) - \delta(X_t) \Big| \mathcal{H}_{t-1} \Bigg]$$

$$= \mathbb{E}\left[\frac{1}{\widehat{\delta}^A_{t-1}(X_t)}(\delta^Y(X_t) - \delta^A(X_t)\widehat{\delta}_{t-1}(X_t)) + \widehat{\delta}_{t-1}(X_t) - \delta(X_t) \Big| \mathcal{H}_{t-1}\right]$$

$$= \mathbb{E}\left[\frac{\delta^A(X_t)}{\widehat{\delta}^A_{t-1}(X_t)}(\delta(X_t) - \widehat{\delta}_{t-1}(X_t)) + \widehat{\delta}_{t-1}(X_t) - \delta(X_t) \Big| \mathcal{H}_{t-1}\right]$$

$$= \mathbb{E}\left[\frac{1}{\widehat{\delta}^A_{t-1}(X_t)}(\delta^A(X_t) - \widehat{\delta}^A_{t-1}(X_t))(\delta(X_t) - \widehat{\delta}_{t-1}(X_t)) \Big| \mathcal{H}_{t-1}\right]$$

$$\leq C\|\widehat{\delta}_{t-1} - \delta\|_2 \|\widehat{\delta}^A_{t-1} - \delta^A\|_2 \qquad \text{(Assumption 3 and Cauchy-Schwarz)}$$

$$= o_p(t^{-1/2})$$

Using a similar argument as in the previous section, we have that $\sqrt{T}\left(\frac{1}{T}\sum_{t=1}^T \Delta^A_t\right) = o_p(1)$.

We now focus on the empirical process term $\Delta^B$. We will show that $\mathbb{E}[\Delta^B] = 0$ and $\text{Var}(\Delta^B) = o_p(1)$ and then apply Chebyshev's inequality to reach the desired conclusion. We now turn to the empirical process term $\Delta^B$. Our goal is to show that $\mathbb{E}[\Delta^B] = 0$ and $\text{Var}(\Delta^B) = o_p(1)$, which together imply that $\Delta^B$ is $o_p(1)$ by Chebyshev's inequality.

Let $\phi_t(\widehat{\eta}_t) := \phi(X_t, Z_t, A_t, Y_t; \pi_t, \widehat{\eta}_t)$ and $\phi_t(\eta) := \phi(X_t, Z_t, A_t, Y_t; \pi_t, \eta)$. We tackle the mean:

$$\mathbb{E}[\Delta^B] = \mathbb{E}\left[\frac{1}{\sqrt{T}}\sum_{t=1}^T (\phi_t(\widehat{\eta}_t) - \phi_t(\eta) - \mathbb{E}[\phi_t(\widehat{\eta}_t) - \phi_t(\eta) \mid \mathcal{H}_{t-1}])\right]$$

$$= \frac{\sqrt{T}}{T}\sum_{t=1}^T (\mathbb{E}[\phi_t(\widehat{\eta}_t) - \phi_t(\eta)] - \mathbb{E}[\phi_t(\widehat{\eta}_t) - \phi_t(\eta)]) = 0. \qquad \text{(Iterated expectations)}$$

Let us now bound $\text{Var}(\Delta^B)$. Since the summands in $\Delta^B$ are conditionally mean-zero and adapted to the filtration $\mathcal{H}_{t-1}$, the cross-terms vanish by martingale difference independence. Thus:

$$\text{Var}(\Delta^B) = \text{Var}\left(\frac{1}{\sqrt{T}}\sum_{t=1}^T (\phi_t(\widehat{\eta}_t) - \phi_t(\eta) - \mathbb{E}[\phi_t(\widehat{\eta}_t) - \phi_t(\eta) \mid \mathcal{H}_{t-1}])\right)$$

$$= \frac{1}{T}\sum_{t=1}^T \text{Var}\left(\phi_t(\widehat{\eta}_t) - \phi_t(\eta) - \mathbb{E}[\phi_t(\widehat{\eta}_t) - \phi_t(\eta) \mid \mathcal{H}_{t-1}]\right)$$

$$= \frac{1}{T}\sum_{t=1}^T \mathbb{E}\left[\text{Var}\left(\phi_t(\widehat{\eta}_t) - \phi_t(\eta) \mid \mathcal{H}_{t-1}\right)\right]$$

$$\leq \frac{1}{T}\sum_{t=1}^T \mathbb{E}\left[\mathbb{E}[(\phi_t(\widehat{\eta}_t) - \phi_t(\eta))^2 \mid \mathcal{H}_{t-1}]\right]$$

where we used the inequality $\text{Var}(X - \mathbb{E}[X \mid \mathcal{F}]) \leq \mathbb{E}[X^2]$ for any square-integrable $X$. We now stochastically bound $\mathbb{E}[(\phi_t(\widehat{\eta}_t) - \phi_t(\eta))^2 \mid \mathcal{H}_{t-1}]$. First, we note:

$$\phi_t(\widehat{\eta}_t) - \phi_t(\eta)$$

$$= \frac{2Z - 1}{Z\pi_t(X_t) + (1-Z)(1 - \pi_t(X_t))} \cdot$$

$$\left\{ Y_t \frac{\delta^A(X_t) - \widehat{\delta}^A_{t-1}(X_t)}{\delta^A(X_t)\widehat{\delta}^A_{t-1}(X_t)} - A_t \left(\frac{\widehat{\delta}_{t-1}(X_t)}{\widehat{\delta}^A_{t-1}(X_t)} - \frac{\delta(X_t)}{\delta^A(X_t)}\right) + \left(\frac{\widehat{\mu}^Y_{t-1}(0, X_t)}{\widehat{\delta}^A_{t-1}(X_t)} - \frac{\mu^Y(0, X_t)}{\delta^A(X_t)}\right)\right.$$

$$\left. + \left(\frac{\widehat{\mu}^A_{t-1}(0, X_t)\widehat{\delta}_{t-1}(0, X_t)}{\widehat{\delta}^A_{t-1}(X_t)} - \frac{\mu^A(0, X_t)\delta(0, X_t)}{\delta^A(X_t)}\right) + (\widehat{\delta}_{t-1}(X_t) - \delta(X_t))\right\}.$$

Then:

$$\mathbb{E}[(\phi_t(\widehat{\eta}_t) - \phi_t(\eta))^2 \mid \mathcal{H}_{t-1}] = \mathbb{E}\left[\mathbb{E}\left[(\phi_t(\widehat{\eta}_t) - \phi_t(\eta))^2 \mid X_t, \mathcal{H}_{t-1}\right]\right]$$

$$\leq 2\mathbb{E}\left[\frac{1}{\pi_t(X_t)}\mathbb{E}[Y_t^2 \mid Z_t = 1, X_t]\frac{(\delta^A(X_t) - \widehat{\delta}^A_{t-1}(X_t))^2}{(\delta^A(X_t)\widehat{\delta}^A_{t-1}(X_t))^2}\Bigg|\mathcal{H}_{t-1}\right] \tag{a}$$

$$+ 2\mathbb{E}\left[\frac{1}{\pi_t(X_t)}\mathbb{E}[A_t^2 \mid Z_t = 1, X_t]\left(\frac{\widehat{\delta}_{t-1}(X_t)}{\widehat{\delta}^A_{t-1}(X_t)} - \frac{\delta(X_t)}{\delta^A(X_t)}\right)^2\Bigg|\mathcal{H}_{t-1}\right] \tag{b}$$

$$+ 2\mathbb{E}\left[\frac{1}{\pi_t(X_t)}\left(\frac{\widehat{\mu}^Y_{t-1}(0, X_t)}{\widehat{\delta}^A_{t-1}(X_t)} - \frac{\mu^Y(0, X_t)}{\delta^A(X_t)}\right)^2\Bigg|\mathcal{H}_{t-1}\right] \tag{c}$$

$$+ 2\mathbb{E}\left[\frac{1}{\pi_t(X_t)}\left(\frac{\widehat{\mu}^A_{t-1}(0, X_t)\widehat{\delta}_{t-1}(0, X_t)}{\widehat{\delta}^A_{t-1}(X_t)} - \frac{\mu^A_{t-1}(0, X_t)\delta_{t-1}(0, X_t)}{\delta^A(X_t)}\right)^2\Bigg|\mathcal{H}_{t-1}\right] \tag{d}$$

$$+ 2\mathbb{E}\left[(\widehat{\delta}_{t-1}(X_t) - \delta(X_t))^2\big|\mathcal{H}_{t-1}\right] \tag{e}$$

$$- 2\mathbb{E}\left[\frac{1}{1 - \pi_t(X_t)}\mathbb{E}[Y_t^2 \mid Z_t = 0, X_t]\frac{(\delta^A(X_t) - \widehat{\delta}^A_{t-1}(X_t))^2}{(\delta^A(X_t)\widehat{\delta}^A_{t-1}(X_t))^2}\Bigg|\mathcal{H}_{t-1}\right] \tag{a}$$

$$- 2\mathbb{E}\left[\frac{1}{1 - \pi_t(X_t)}\mathbb{E}[A_t^2 \mid Z_t = 1, X_t]\left(\frac{\widehat{\delta}_{t-1}(X_t)}{\widehat{\delta}^A_{t-1}(X_t)} - \frac{\delta(X_t)}{\delta^A(X_t)}\right)^2\Bigg|\mathcal{H}_{t-1}\right] \tag{b}$$

$$- 2\mathbb{E}\left[\frac{1}{1 - \pi_t(X_t)}\left(\frac{\widehat{\mu}^Y_{t-1}(0, X_t)}{\widehat{\delta}^A_{t-1}(X_t)} - \frac{\mu^Y(0, X_t)}{\delta^A(X_t)}\right)^2\Bigg|\mathcal{H}_{t-1}\right] \tag{c}$$

$$- 2\mathbb{E}\left[\frac{1}{1 - \pi_t(X_t)}\left(\frac{\widehat{\mu}^A_{t-1}(0, X_t)\widehat{\delta}_{t-1}(0, X_t)}{\widehat{\delta}^A_{t-1}(X_t)} - \frac{\mu^A_{t-1}(0, X_t)\delta_{t-1}(0, X_t)}{\delta^A(X_t)}\right)^2\Bigg|\mathcal{H}_{t-1}\right] \tag{d}$$

$$- 2\mathbb{E}\left[(\widehat{\delta}_{t-1}(X_t) - \delta(X_t))^2\big|\mathcal{H}_{t-1}\right] \tag{e}$$

Bounding all terms using Assumption 1 and Assumption 3, we have:

$$\mathbb{E}[(\phi_t(\widehat{\eta}_t) - \phi_t(\eta))^2 \mid \mathcal{H}_{t-1}]$$

$$\leq 4k_t C^4 \epsilon^2_{\delta^A} \|\widehat{\delta}^A_{t-1} - \delta^A\|_2^2 \tag{a}$$

$$+ 8k_t C^2 \epsilon^2_{\delta^A}(\|\widehat{\delta}_{t-1} - \delta\|_2^2 + 4C^2\|\widehat{\delta}^A_{t-1} - \delta^A\|_2^2) \tag{b}$$

$$+ 8k_t C^2 \epsilon^2_{\delta^A}(\|\widehat{\mu}^Y_{t-1}(0, \cdot) - \mu^Y(0, \cdot)\|_2^2 + C^2\|\widehat{\delta}^A_{t-1} - \delta^A\|_2^2) \tag{c}$$

$$+ 8k_t C^2 \epsilon^2_{\delta^A}(4C^4\|\widehat{\mu}^A_{t-1}(0, \cdot) - \mu^A(0, \cdot)\|_2^2 + 4C^4\|\widehat{\delta}^A_{t-1} - \delta^A\|_2^2 + C^2\|\widehat{\delta}_{t-1} - \delta\|_2^2) \tag{d}$$

$$+ \|\widehat{\delta}_{t-1}(X_t) - \delta(X_t)\|_2^2 \tag{e}$$

$$= o_p(1)$$

where the last line follows from the fact that (a-e) are $o_p(1)$ from the premise of Theorem 3. Since each term $\mathbb{E}\left[(\phi_t(\widehat{\eta}_t) - \phi_t(\eta))^2 \mid \mathcal{H}_{t-1}\right]$ is nonnegative, uniformly bounded, and satisfies $o_p(1)$, it follows by Cesàro averaging that $\frac{1}{T}\sum_{t=1}^T \mathbb{E}\left[(\phi_t(\widehat{\eta}_t) - \phi_t(\eta))^2 \mid \mathcal{H}_{t-1}\right] = o_p(1)$. Thus, $\text{Var}(\Delta^B) = o_p(1)$ and we can apply Chebyshev's inequality to obtain $P(|\Delta^B| \geq \varepsilon) \leq \text{Var}(\Delta^B)/\varepsilon^2, \forall \varepsilon > 0$. Therefore, $\Delta^B$ is $o_p(1)$, as desired. Putting everything together, we conclude that $\sqrt{T}\left(\frac{1}{T}\sum_{t=1}^T m_t\right) = o_p(1)$ and the conclusion of Theorem 3 holds.

# G  Proof of Theorem 4 and Corollary 5

Letting $\phi_t(\pi, \eta) := \phi(X_t, Z_t, A_t, Y_t; \pi, \eta)$ for any $\pi, \eta$ and , we decompose $\widehat{\tau}_T^{\mathrm{AMRIV}} - \tau$ as follows:

$$
\begin{aligned}
\widehat{\tau}_T^{\mathrm{AMRIV}} - \tau &= \frac{1}{T} \sum_{t=1}^{T} \phi_t(\pi_t, \widehat{\eta}_t) - \tau \\
&= \underbrace{\frac{1}{T} \sum_{t=1}^{T} \left( \phi_t(\pi_t, \widehat{\eta}_t) - \mathbb{E}[\phi_t(\pi_t, \widehat{\eta}_t) \mid \mathcal{H}_{t-1}] \right)}_{\Delta^A} + \underbrace{\frac{1}{T} \sum_{t=1}^{T} \left( \mathbb{E}[\phi_t(\pi_t, \widehat{\eta}_t) \mid \mathcal{H}_{t-1}] - \tau \right)}_{\Delta^B}
\end{aligned}
$$

We will now show that $\Delta^A$ is $O_p(T^{-1/2})$ via a similar argument as in Appendix F and $\Delta^B$ is $O_p\left( \|\widehat{\delta}_T^A - \delta^A\|_2 \|\widehat{\delta}_T - \delta\|_2 \right)$.

Write $\Delta_t^A := \phi_t(\pi_t, \widehat{\eta}_t) - \mathbb{E}[\phi_t(\pi_t, \widehat{\eta}_t) \mid \mathcal{H}_{t-1}]$ and note that $\Delta_t^A$ is an MDS by construction, *i.e.* $\mathbb{E}[\Delta_t^A \mid \mathcal{H}_{t-1}] = 0$. Let $\widetilde{V}(\pi) := \mathrm{Var}_\pi(\phi_t(\pi, \tilde{\eta}))$ where $\mathrm{Var}_\pi$ indicates the variance over data where $Z \sim \mathrm{Bern}(\pi(X_t))$. Thus, it suffices to show that $\mathbb{E}[(\Delta_t^A)^2 \mid \mathcal{H}_{t-1}] \xrightarrow{p} \sigma^2$, where $\sigma^2 = \widetilde{V}(\pi)$. Then, the result follows by tracing the rest of the proof in Appendix F.

Write $\Lambda_t := \phi_t(\pi_t, \widehat{\eta}_t) - \phi_t(\pi_t, \widetilde{\eta})$ and note

$$
\mathrm{Var}\big(\phi_t(\pi_t, \widehat{\eta}_t) \mid \mathcal{H}_{t-1}\big) = \mathrm{Var}\big(\phi_t(\pi_t, \tilde{\eta}) \mid \mathcal{H}_{t-1}\big) + \mathrm{Var}(\Lambda_t \mid \mathcal{H}_{t-1}) + 2\,\mathrm{Cov}\big(\phi_t(\tilde{\eta}), \Delta_t \mid \mathcal{H}_{t-1}\big)
$$

and thus

$$
\begin{aligned}
\big| \mathrm{Var}\big(\phi_t(\pi_t, \widehat{\eta}_t) \mid \mathcal{H}_{t-1}\big) &- \mathrm{Var}\big(\phi_t(\pi_t, \tilde{\eta}) \mid \mathcal{H}_{t-1}\big) \big| \\
&\leq \mathrm{Var}(\Lambda_t \mid \mathcal{H}_{t-1}) + 2\sqrt{\mathrm{Var}(\Lambda_t \mid \mathcal{H}_{t-1})\,\mathrm{Var}(\phi_t(\pi_t, \widetilde{\eta}) \mid \mathcal{H}_{t-1})}
\end{aligned}
$$

Since, $\mathrm{Var}(\phi_t(\pi_t, \widetilde{\eta}) \mid \mathcal{H}_{t-1})$ is bounded by Assumption 3, we just need to show that $\mathrm{Var}(\Lambda_t \mid \mathcal{H}_{t-1}) = o_p(1)$:

$$
\begin{aligned}
\mathrm{Var}(\phi_t(\pi_t, \widetilde{\eta}) \mid \mathcal{H}_{t-1}) &\leq \mathbb{E}\left[ (\phi_t(\pi_t, \widehat{\eta}_t) - \phi_t(\pi_t, \widetilde{\eta}))^2 \mid \mathcal{H}_{t-1} \right] \\
&\leq \tilde{C} k_t \big( \|\widehat{\delta}_{t-1} - \tilde{\delta}\|_2^2 + \|\widehat{\mu}_{t-1}^Y(0, \cdot) - \tilde{\mu}^Y(0, \cdot)\|_2^2 + \|\widehat{\mu}_{t-1}^A(0, \cdot) - \tilde{\mu}^A(0, \cdot)\|_2^2 + \|\widehat{\delta}_{t-1}^A - \tilde{\delta}^A\|_2^2 \big) \\
&\qquad\qquad\qquad\qquad\qquad\qquad\qquad\qquad\qquad\qquad\qquad\qquad\text{(Parallelogram law)} \\
&= o_p(1) \qquad\qquad\qquad\qquad\qquad\qquad\qquad\qquad\qquad\qquad\text{(Theorem assumptions)}
\end{aligned}
$$

where $\tilde{C}$ encompasses the constants $\epsilon$ and $C$ from Assumption 3 and the theorem's premise. Thus, since $\mathrm{Var}\big(\phi_t(\pi_t, \widehat{\eta}_t) \mid \mathcal{H}_{t-1}\big) \xrightarrow{p} \mathrm{Var}\big(\phi_t(\pi_t, \widetilde{\eta}) \mid \mathcal{H}_{t-1}\big) \xrightarrow{p} \widetilde{V}_\pi$, we can use Theorem 8 from Appendix F and retrace the same arguments to obtain $\Delta^A = O_p(T^{-1/2})$ due to the Martingale CLT.

Now, we study $\Delta^B$:

$$
\begin{aligned}
&\Delta_t^B \\
&= \mathbb{E}[\phi_t(\pi_t, \widehat{\eta}_t) \mid \mathcal{H}_{t-1}] - \mathbb{E}[\delta(X)] \\
&= \mathbb{E}[\mathbb{E}[\phi(X_t, Z_t, A_t, Y_t; \pi_t, \widehat{\eta}_t) \mid X_t, \mathcal{H}_{t-1}] \mid \mathcal{H}_{t-1}] - \mathbb{E}[\delta(X_t) \mid \mathcal{H}_{t-1}] \\
&= \mathbb{E}\big[ \mathbb{E}[\phi(X_t, Z_t, A_t, Y_t; \pi_t, \widehat{\eta}_t) \mid Z_t = 1, X_t, \mathcal{H}_{t-1}] \pi_t(X_t \mid \mathcal{H}_{t-1}) \mid \mathcal{H}_{t-1} \big] \\
&\quad + \mathbb{E}\big[ \mathbb{E}[\phi(X_t, Z_t, A_t, Y_t; \pi_t, \widehat{\eta}_t) \mid Z_t = 0, X_t, \mathcal{H}_{t-1}] (1 - \pi_t(X_t \mid \mathcal{H}_{t-1})) \mid \mathcal{H}_{t-1} \big] \\
&\quad - \mathbb{E}[\delta(X_t) \mid \mathcal{H}_{t-1}] \\
&= \mathbb{E}\Bigg[ \frac{1}{\widehat{\delta}_{t-1}^A(X_t)} \big( \mu^Y(1, X_t) - \mu^A(1, X_t) \widehat{\delta}_{t-1}(X_t) + \widehat{\mu}_{t-1}^Y(0, X_t) - \widehat{\mu}_{t-1}^A(0, X_t) \widehat{\delta}_{t-1}(X_t) \big) \\
&\qquad - \frac{1}{\widehat{\delta}_{t-1}^A(X_t)} \big( \mu^Y(0, X_t) - \mu^A(0, X_t) \widehat{\delta}_{t-1}(X_t) + \widehat{\mu}_{t-1}^Y(0, X_t) - \widehat{\mu}_{t-1}^A(0, X_t) \widehat{\delta}_{t-1}(X_t) \big) \\
&\qquad + \widehat{\delta}_{t-1}(X_t) - \delta_{t-1}(X_t) \Big| \mathcal{H}_{t-1} \Bigg]
\end{aligned}
$$

$$= \mathbb{E}\left[\frac{1}{\widehat{\delta}^A_{t-1}(X_t)}(\delta^Y(X_t) - \delta^A(X_t)\widehat{\delta}_{t-1}(X_t)) + \widehat{\delta}_{t-1}(X_t) - \delta_{t-1}(X_t)\Big|\mathcal{H}_{t-1}\right]$$

$$= \mathbb{E}\left[\frac{\delta^A(X_t)}{\widehat{\delta}^A_{t-1}(X_t)}(\delta(X_t) - \widehat{\delta}_{t-1}(X_t)) + \widehat{\delta}_{t-1}(X_t) - \delta_{t-1}(X_t)\Big|\mathcal{H}_{t-1}\right]$$

$$= \mathbb{E}\left[\frac{1}{\widehat{\delta}^A_{t-1}(X_t)}(\delta^A(X_t) - \widehat{\delta}^A_{t-1}(X_t))(\delta(X_t) - \widehat{\delta}_{t-1}(X_t))\Big|\mathcal{H}_{t-1}\right]$$

$$\leq C\|\widehat{\delta}_{t-1} - \delta\|_2\|\widehat{\delta}^A_{t-1} - \delta^A\|_2 \qquad \text{(Assumption 3 and Cauchy-Schwarz)}$$

Thus, $\Delta^B_t$ is $O_p(\|\widehat{\delta}_T - \delta\|_2\|\widehat{\delta}^A_T - \delta^A\|_2)$, as desired. Corollary 5 follows immediately by noting that if either $\widehat{\delta}_{t-1}$ or $\widehat{\delta}^A_{t-1}$ are consistent(*i.e.* $o_p(1)$), then $\Delta^B$ is also consistent and $|\tau^{\text{AMRIV}}_T - \tau| = o_p(1)$.

## H    Experimental Details

This appendix provides additional details for the simulation experiments described in Section 7, including exact hyperparameters, model components, and execution setup. All experiments were run on a Perlmutter compute node with 256 CPU cores at the National Energy Research Scientific Computing Center (NERSC) [37] and required approximately 40–50 minutes per configuration. Random Forests were implemented using `scikit-learn` [44], and parallelization was handled via `joblib`. Full code for generating data, running experiments, and reproducing all figures is available at `https://github.com/CausalML/Adaptive-IV`, with instructions in the `README.md`.

Each estimator was evaluated on 1000 independent synthetic trials. Simulations were run over $T = 2000$ rounds with a $T_0 = 200$ burn-in period, and nuisance estimators were updated in mini-batches of 200. For all adaptive methods, we applied the truncated optimal allocation policy from Eq. (7), with a truncation schedule $k_t = 2/0.999^t$. Oracle methods used ground-truth nuisance functions, while misspecified estimators were constructed by replacing $\mu^Y(1, X)$ with a constant regressor fit to the average oracle value.

Unless otherwise stated, outcome and residual variance functions were modeled via Random Forests with 100 trees, maximum depth 5, and minimum leaf size 5. The compliance model $\mu^A(1, X)$ was learned with a shallower forest (depth 3, minimum leaf size 30), and $\mu^A(0, X)$ was zero by construction due to one-sided noncompliance. For the A2IPW estimator, we followed Kato et al. [27] and estimated outcome means and second moments using random forests (depth 5, leaf size 100) and used a Neyman-style allocation based on observed outcomes. All figures report results averaged over replicates, with confidence intervals based on empirical standard errors.

### H.1    Simulation Studies with Synthetic Data

We generate the data sequentially for each time $t \in [1, T + T_0]$ using the following one-sided noncompliance setup:

$$X_t \sim \text{Unif}(0, 2)^d, \quad Z_t \sim \text{Bern}(\pi_t(X_t \mid \mathcal{H}_{t-1})) \qquad \text{(Covariates \& Instrument)}$$

$$\mu^A(0, X_t) = 0, \quad \delta^A(X_t) = \mu^A(1, X_t) = \sigma(2X_t[1]) \qquad \text{(Compliance Scores)}$$

$$C_t \sim \text{Bern}(\delta^A(X_t)), \quad A_t = C_t \cdot Z_t \qquad \text{(Treatment Assignment)}$$

$$U_t = u(1 - C_t) \qquad \text{(Unobserved Confounder)}$$

$$Y_t = f(A_t, X_t) + U_t + \epsilon_{A_t}, \quad \epsilon_{A_t} \sim \text{Unif}[-g(A_t, X_t), g(A_t, X_t)] \qquad \text{(Outcome Function)}$$

where $T_0$ is the burn-in period, $C_t$ is the (unknown) compliance indicator, $\sigma$ is the logistic sigmoid, Unif and Bern are the uniform and the Bernoulli distributions, respectively. We utilize the following instantiations for $d, u, f, g$:

$$d = 5$$
$$u = -2.0$$
$$f(A, X) = 1 + A + X[1] + 2a\,(X^\top\beta) + 0.75a\,X[1]^2$$
$$g(A, X) = \sqrt{3 \cdot (v_1 \cdot A + (v_0 \cdot X[1] + v_1) \cdot (1 - A))}, \quad v_0 = 4.0, v_1 = 0.25$$

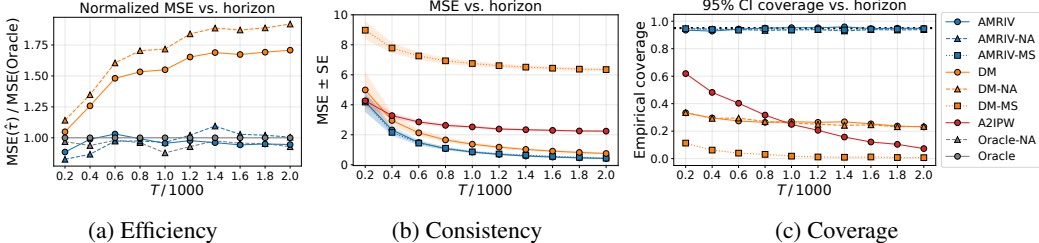

Figure 4: Performance of different estimators on TripAdvisor simulated data. **(a)** Efficiency: Normalized MSE versus an oracle benchmark. **(b)** Consistency: MSE $\pm$ standard error. **(c)** Coverage: Empirical coverage of 95% confidence intervals.

where $X[1]$ denotes the first coordinate of the covariate vector $X \in \mathbb{R}^d$ and $\beta \in \mathbb{R}^d, \beta \sim$ Unif$[-1, 1]^d$ is a parameter vector that is fixed over the 1000 simulations (we used a seed of 1 for reproducibility purposes.). $g(A, X)$ was chosen such that $\text{Var}(\epsilon_A \mid X) = v_1 \cdot A + (v_0 \cdot X[1] + v_1) \cdot (A - 1)$ where $v_0, v_1$ are constants.

## H.2 Simulation Studies with Semi-Synthetic Data

To complement our synthetic evaluation, we conduct additional experiments using a semi-synthetic setting derived from a real-world dataset collected by TripAdvisor. The original data-generating process (DGP) was introduced by Syrgkanis et al. [52] and is publicly available on GitHub. In the original A/B test, users were randomly assigned to one of two groups: group A (instrument $Z = 1$) was offered a simplified membership sign-up experience, while group B ($Z = 0$) saw the default interface. This encouragement increased the likelihood of signing up for a membership (treatment $A$), though actual uptake remained endogenous due to user-specific factors.

The covariates $X \in \mathbb{R}^{10}$ capture rich pre-treatment user behavior and demographics. These include: prior platform revenue, visit frequencies to different TripAdvisor sections (hotels, restaurants, experiences, flights, and vacation rentals) over a 28-day pre-experimental window, engagement through free channels (e.g., email), locale information, and operating system type. The binary treatment $A$ indicates whether the user became a member during the experiment, while the outcome $Y$ records the total number of days the user visited TripAdvisor during the study period.

We preserve the original covariate structure and instrument assignment mechanism, but modify the outcome model to introduce heteroskedasticity by adding log-normal noise whose variance depends on treatment status. This choice reflects the heavy-tailed nature of usage metrics in online platforms [5, 31], and results in a more realistic and challenging estimation task. The full data-generating process is provided below ("*" indicates same as original).

**TripAdvisor Data-Generating Process.** We simulate tuples $(X, A(0), A(1), Y(0), Y(1))$ via:

$$X \sim \text{TripAdvisor pre-treatment covariates}, \qquad (*)$$
$$\nu \sim \text{Unif}[-5, 5], \qquad (\text{latent user heterogeneity}^*)$$
$$A(1) \sim \text{Bernoulli}\left(0.8 \cdot \sigma(0.4X_1 + \nu)\right), \quad A(0) \sim \text{Bernoulli}(0.006), \qquad (\text{compliance}^*)$$
$$\varepsilon_1 \sim \text{LogNormal}(0, \sigma_1), \quad \varepsilon_0 \sim \text{LogNormal}(0, \sigma_0), \qquad (\textbf{new: } \text{heavy-tailed errors})$$
$$Y(1) = f(X) + 2\nu + 5 \cdot \mathbb{I}[X[1] > 0] + \varepsilon_1, \qquad (\text{potential outcome for } A = 1^*)$$
$$Y(0) = 2\nu + 5 \cdot \mathbb{I}[X[1] > 0] + \varepsilon_0, \qquad (\text{potential outcome for } A = 0^*)$$

where $\sigma_0 = 1.5$ and $\sigma_1 = 0.25$, and the structural CATE function is defined as:

$$f(X) = 0.8 + 0.5 \cdot \phi(X_1) - 3.0X[7],$$

with $\phi(X_1) := 5 \cdot \mathbb{I}[X[1] > 5] + 10 \cdot \mathbb{I}[X_1 > 15] + 5 \cdot \mathbb{I}[X_1 > 20]$.

We illustrate our results on Figure 4. To maintain readability in the plots, we define the misspecified outcome model $\widehat{\mu}^Y(1, X)$ as the oracle $\mu^Y(1, X)$ plus a constant shift.

**Adaptivity.** As shown in panel (a), adaptive allocation improves the efficiency of both the DM and AMRIV estimators (particularly at larger $T$), with AMRIV again approaching the oracle benchmark. Interestingly, AMRIV, AMRIV-NA, and Oracle-NA slightly outperform the fully adaptive Oracle

in some regimes—likely due to extreme compliance scores in this DGP ($\delta^A(X) \to 0$), which inflate the asymptotic variance when using oracle denominators. In contrast, estimators with learned denominators can perform better in finite samples [28, 51]. This also helps explain the narrower variance gap between AMRIV and AMRIV-NA relative to the synthetic setting, as the variance is dominated by low-compliance regions rather than outcome variance between instrument arms.

**Consistency.** Panel (b) confirms that AMRIV, AMRIV-NA, and DM all converge to the true $\tau$, with AMRIV variants consistently achieving lower error. As expected, A2IPW fails to converge due to uncorrected confounding, while DM-MS diverges due to misspecification of $\delta(X)$. In contrast, AMRIV-MS remains consistent—further validating the multiply-robust guarantee from Theorem 4.

**Coverage.** Panel (c) shows that AMRIV, AMRIV-NA, and AMRIV-MS maintain valid 95% confidence interval coverage, consistent with the asymptotic normality result in Theorem 3. All other estimators—including DM-MS and A2IPW—under-cover severely as $T$ increases, reflecting bias under misspecification or confounding.

# I   Limitations and Broader Impacts

## Limitations

While AMRIV is grounded in semiparametric theory and achieves strong empirical performance, there are several limitations we highlight. First, our method relies on standard IV identification assumptions (Assumption 1) and the unconfounded compliance assumption (Assumption 2), which—while weaker than ignorability—are still untestable and may be violated in practice. In particular, the exclusion restriction and the unconfounded compliance assumption may not hold even in observational settings where the instrument is randomized. Second, AMRIV assumes access to flexible, sequentially consistent nuisance estimators, which may be difficult to train or tune in low-data regimes or in the presence of heavy-tailed outcomes. Third, our analysis focuses on a binary instrument and binary treatment; extending the framework to multi-valued or continuous instruments remains an open challenge.

## Broader Impacts

This work contributes to the growing intersection of causal inference and adaptive experimentation, enabling more data-efficient and statistically principled estimation in settings with noncompliance. Potential applications include health interventions and online recommendation systems, where experimenters can encourage behavior but not enforce it. AMRIV allows experimenters to make better use of limited resources while supporting robust inference under endogenous treatment selection under unobserved confounding. However, we caution that the validity of conclusions drawn from AMRIV hinges on the identification assumptions and data quality. In high-stakes settings, particularly those involving marginalized or vulnerable populations, improper use or misinterpretation could lead to harmful decisions. We strongly recommend pairing AMRIV with domain expertise, sensitivity analysis, and uncertainty quantification to ensure responsible deployment and interpretation.

