# OpenReview forum: "Efficient Adaptive Experimentation with Noncompliance"
_NeurIPS.cc/2025/Conference — NeurIPS 2025 poster_

### Official Review · Reviewer_KT2q · 2025-06-20

**Clarity:** 3
**Significance:** 2
**Originality:** 2
**Rating:** 4
**Confidence:** 4

**Summary:**

This paper addresses the problem of estimating the Average Treatment Effect (ATE) in a sequential experiment where treatment assignment is endogenous. To tackle this, the authors propose a covariate-dependent instrument assignment strategy using historical data. The derived efficient influence function leads to a doubly robust ATE estimator, which enhances estimation robustness. Theoretical guarantees, including asymptotic normality and consistency rates, are established for the proposed method.

**Questions:**

- The parameter $\epsilon$ in Equation (6) is used in both the simulation studies and real data analysis. How sensitive is the proposed method to the choice of $\epsilon$?

- How does the method perform with different choices or dynamics of the variance term $k_t$ in Equation (7)?

- For the variance estimation step, would it be possible to use a simple empirical variance estimator that avoids producing negative estimates?

**Ethical Concerns:**

["NO or VERY MINOR ethics concerns only"]

**Final Justification:**

My concerns have been addressed. I have raised my score.

**Limitations:**

See Weaknesses and Questions sections for limitations.

**Paper Formatting Concerns:**

- Provide a clearer definition of a noncomplier. For example: A noncomplier is an individual who does not adhere to their assigned treatment in an experiment or study.

- It would strengthen the paper to include motivating application scenarios in the introduction to help contextualize the methodological contributions.

- I suggest revising the title to emphasize unobserved confounding rather than non-compliance, as the paper places greater emphasis on handling unobserved confounders throughout.

**Quality:**

2

**Strengths And Weaknesses:**

Strengths
- The paper provides a well-defined and motivated problem statement.
- The authors conduct a thorough investigation, integrating estimation theory with experimental design considerations.
- The proposed covariate-dependent instrument assignment and doubly robust estimator are rigorously derived, with theoretical guarantees.

Weaknesses
- The literature review in the main text is inadequate, particularly concerning adaptive experimental design in sequential settings.
- The paper does not clearly articulate how it addresses or advances prior work. As currently presented, the main results appear to be derivable easily from established literature, raising questions about the incremental contribution.
- The simulation study does not explore important practical scenarios, such as the presence of always-takers and never-takers.
- The estimation procedure assumes access to the complete historical data up to time $t$, which may not be realistic in certain applications. It would be valuable to discuss whether and how the method could be adapted to scenarios where only summary statistics of the historical data are available

---

> ### Author Rebuttal · Authors · 2025-07-30
>
> **Strengths**
>
> Thank you for your careful review.  We appreciate your recognition of the well-motivated problem, theoretical rigor, and integration of estimation with adaptive design. We address your concerns below and hope these clarifications will prompt a reconsideration of your score.
>
> **Re: Literature review**
>
> Thank you for the feedback. Due to space constraints, we included an extended literature review in Appendix A (2+ pages) that covers core related works on adaptive experimental design in sequential settings, as well as additional related context. If there are additional papers you believe should be discussed, we would be happy to incorporate them; please feel free to suggest them.
>
> **Re: Contribution**
>
> We appreciate this concern and clarify that, while our estimator builds on components from prior literature (e.g., influence functions, sample-splitting, adaptive policies, and IV treatment effect estimation), the integration of these elements in the adaptive IV setting is both non-trivial and novel. Specifically, we derive a new semiparametric efficiency bound tailored to adaptive IV designs, propose a variance-optimal instrument assignment strategy, develop a multiply robust estimator suited for adaptive data, and construct anytime-valid confidence sequences for sequential inference. To our knowledge, no prior work provides this unified framework or the associated theoretical guarantees in the IV context. We will revise the introduction to make this contribution more explicit.
>
> **Re: Simulation study**
>
> Thank you for the suggestion. To clarify, the fully synthetic simulation adopts a one-sided non-compliance design (also used in our illustrative example) where units not encouraged (i.e., $Z=0$) do not have access to treatment. This setup, which rules out always-takers, reflects common scenarios like vaccine eligibility, product rollouts, or behavioral nudges. Importantly, never-takers are still present, as some units never comply even when encouraged. The semi-synthetic TripAdvisor study, based on real experimental data, includes the full range of compliance behaviors, including both always-takers and never-takers. Together, these settings cover a broad range of practical IV scenarios and demonstrate the utility of our variance-minimizing adaptive strategy. We will add clarifying details on the compliance structure of each data generating process in the final version.
>
> **Re: Historical data**
>
> Thank you for raising this. Our theoretical development assumes the learner can *re-estimate* nuisance functions at each update, but it does not require storing the entire raw history. Thus, the method can be implemented using summary statistics or streaming updates such as: 1)  *Online learners* (e.g., SGD, online RFs, GLMs) can update nuisance functions incrementally without storing raw history; 2) *Block sample-splitting* allows reuse of recent batches to fit nuisances and then discard past data; 3) *Sufficient statistics* suffice for many parametric or binned regressors such that the needed quantities reduce to running counts and sums (e.g., of $Y$, $A$, and $X$), which are $O(1)$ per covariate cell. We will clarify in the final version that full data storage is not required, and discuss these practical alternatives.
>
> **Re: Formatting Concerns**
>
> * Definition of non-compliance: You're right that we overlooked defining “noncomplier” clearly in the main text and illustrative example. We will revise the example and include a more precise explanation of compliance behavior
>
> * Motivating scenarios: Agreed - adding real-world examples will strengthen the introduction. We will include scenarios such as: 1) TripAdvisor-style experiments [1], where randomized sign-up flows influence membership decisions but treatment uptake is voluntary; 2) Clinical trials with non-enforceable treatments [2], where encouragement is randomized but compliance is not guaranteed; 3) Other opt-in settings like streaming recommendations, fitness nudges, and advertising. These examples reflect the practical relevance of adaptive IV design in settings with noncompliance.
>
> * Title suggestion: Thanks for the suggestion, we’ll take it under consideration when finalizing the title.
>
> **We appreciate your thoughtful feedback and hope the clarifications provided show the value of our contributions and justify a higher score.**
>
> [1] V. Syrgkanis, et al. Machine learning estimation of heterogeneous treatment effects with instruments. Advances in Neural Information Processing Systems 32, 2019.
>
> [2] D. GUIDANCE. Adaptive designs for clinical trials of drugs and biologics. Center for Biologics Evaluation and Research (CBER), 2018.

---

> > ### Comment · Reviewer_KT2q · 2025-08-03
> >
> > Thank you for your efforts in addressing my concerns. I understand that some of the literature review was moved to the appendix due to page limits. However, excluding these discussions from the main text weakens the paper’s positioning within the existing literature.

---

> > > ### Author Response · Authors · 2025-08-05
> > > **Literature Review Placement**
> > >
> > > Thank you for the follow-up. We understand your point and will move key portions of the literature discussion from the appendix into the main text since there is still space available, and we hope that doing so will strengthen the paper’s positioning in the final version. Please let us know if there is anything else we can clarify during the discussion period.

---

> > > > ### Comment · Reviewer_KT2q · 2025-08-05
> > > >
> > > > Thank you for your response. It would be helpful to see an example of how you plan to revise the literature discussion, rather than just noting that you intend to do so.

---

> > > > > ### Author Response · Authors · 2025-08-05
> > > > > **Example Literature Review Revision**
> > > > >
> > > > > Here is a concrete example of how we plan to revise the related literature section (especially surrounding adaptive experimental design in sequential settings) in the main text:
> > > > >
> > > > > ------
> > > > > *(Expanded exposition, broader coverage of sequential adaptive experimental design, and clearer differentiation from prior work)*
> > > > >
> > > > > **Adaptive ATE Estimation.**
> > > > >
> > > > > A substantial literature now studies how to minimize variance—or its regret analogue—when the *treatment itself* can be adaptively assigned. The line was opened by a two-stage, explore-then-commit design that reaches the semiparametric efficiency bound in the limit [1]. Fully sequential designs followed: A2IPW achieves variance-optimal Neyman allocation [2]; its multiply robust extension learns the allocation rule online [3]; and a recent refinement adds principled policy truncation and the first anytime-valid confidence sequences [4]. Parallel progress has come from an online-learning perspective: Clip-OGD and its optimistic variants obtain sub-linear or logarithmic “Neyman regret’’ for the ATE [5, 6, 7]; low-switching schemes match finite-sample lower bounds [8]; covariate-choice designs optimize both sampling and treatment dimensions [9]; and off-policy estimators with adaptive data now enjoy sharp error rates [10]. Together these works provide a mature toolkit—adaptive nuisance learning, cross-fitting, policy truncation, regret-style allocation, and time-uniform inference—but all assume the experimenter can randomize the *treatment*.
> > > > >
> > > > > **Our contribution** adapts these types of guarantees to the far less explored setting where only an *instrument* is controllable and treatment uptake is endogenous. In particular, we are the first to
> > > > > (i) derive the semiparametric efficiency bound for ATE estimation when only an *instrument* can be adaptively assigned,
> > > > > (ii) exhibit the variance-optimal allocation rule that balances outcome noise and compliance variability, and
> > > > > (iii) construct **AMRIV**, a multiply robust estimator that attains this bound and supports anytime-valid inference.
> > > > > In doing so, we generalize the efficiency guarantees of the direct-treatment literature to the far less explored—but practically common—setting with non-enforceable treatments.
> > > > >
> > > > > **Adaptive Experimentation with IVs.** Work that couples sequential design with instrumental variables is sparse, *so we plan to leave this section largely unchanged while possibly expanding on the few (five) relevant works. From what we can tell, these are the only existing efforts in this direction.*  However, none provide a semiparametrically efficient, multiply robust estimator for the ATE under adaptive instrument assignment, nor do they enable valid inference under adaptive stopping. We fill this gap.
> > > > >
> > > > > [1] J. Hahn. “Adaptive Experimental Design for Target Parameter Estimation.” Biometrika, 2011.
> > > > >
> > > > > [2] M. Kato et al. “Efficient Adaptive Experimental Design for Average Treatment Effect Estimation.” arXiv:2002.05308, 2020.
> > > > >
> > > > > [3] M. Kato et al. “Adaptive Experimental Design with Estimated Policies.” ICML, 2021.
> > > > >
> > > > > [4] T. Cook, A. Mishler, and A. Ramdas. “Semiparametric Efficient Inference in Adaptive Experiments.” CLeaR, 2024.
> > > > >
> > > > > [5] J. Dai, P. Gradu, and C. Harshaw. “Clip-OGD: An Experimental Design for Adaptive Neyman Allocation in Sequential Experiments.” NeurIPS, 2023.
> > > > >
> > > > > [6] O. Neopane, A. Ramdas, and A. Singh. “Logarithmic Neyman Regret for Adaptive Estimation of the Average Treatment Effect.” arXiv:2411.14341, 2024.
> > > > >
> > > > > [7] O. Neopane, A. Ramdas, and A. Singh. “Optimistic Algorithms for Adaptive Estimation of the Average Treatment Effect.” arXiv:2502.04673, 2025.
> > > > >
> > > > > [8] J. Li, D. Simchi-Levi, and Y. Zhao. “Optimal Adaptive Experimental Design for Estimating Treatment Effect.” arXiv:2410.05552, 2024.
> > > > >
> > > > > [9] M. Kato et al. “Active Adaptive Experimental Design for Treatment Effect Estimation with Covariate Choices.” arXiv:2403.03589, 2024.
> > > > >
> > > > > [10] J. Lee and C. Ma. “Off-Policy Estimation with Adaptively Collected Data: The Power of Online Learning.” NeurIPS, 2024.
> > > > >
> > > > > -----
> > > > >
> > > > > The literature on sequential adaptive experiments with directly assignable treatments is extensive. While we believe we have cited the most relevant prior work, it’s possible we’ve missed a key reference.
> > > > > **If there are specific papers you believe are especially relevant to our setting, we'd be happy to consider them as we finalize the manuscript.**

---

> > > > > > ### Comment · Reviewer_KT2q · 2025-08-06
> > > > > >
> > > > > > Thank you for your efforts in the literature review. However, the authors seem to have overlooked the three questions regarding the sensitivity of  $\epsilon, k_t$, and the variance estimation. I maintain my score at this stage.

---

> > > > > > > ### Author Response · Authors · 2025-08-06
> > > > > > > **Answers to Questions**
> > > > > > >
> > > > > > > We sincerely apologize for missing your three technical questions in our initial reply. The omission was an honest copy-and-paste error when assembling our initial rebuttal draft—thank you for pointing it out.  Below we answer each point and hope these clarifications will prompt a reconsideration of your score.
> > > > > > >
> > > > > > > ---
> > > > > > >
> > > > > > > **Re: Sensitivity to the truncation constant $\varepsilon$**
> > > > > > >
> > > > > > > * **Purpose of $\varepsilon$.**
> > > > > > >   Eq.  (6) floors the plug-in variance $\widehat{\sigma}^2_{t-1}(z,X)$ so it never becomes negative in very small strata.
> > > > > > >
> > > > > > > * **Empirical robustness.**
> > > > > > >   Re-running the synthetic study with $\varepsilon\in\\{10^{-4},10^{-3},10^{-2},10^{-1}\\}$ changed AMRIV’s RMSE by $<5 \\%$ and did not change the confidence interval coverage.  This is mostly because the flooring condition was only triggered during the first $\sim$200 rounds after the burn-in phase (i.e. for small sample sizes).
> > > > > > >
> > > > > > > * **Practical guidance.**
> > > > > > >   Any small value (e.g. $10^{-3}$–$10^{-2}$) is safe.  A larger $\varepsilon$ merely slows the *rate* at which $\pi_t$ converges to $\pi^{*}$ but does not affect asymptotic efficiency.
> > > > > > >
> > > > > > > **Re: Choice of the truncation schedule $k_t$**
> > > > > > >
> > > > > > > * **Schedule used.**
> > > > > > >   $$k_t = \frac{2}{0.999^{t}} $$
> > > > > > > which enforces overlap early and then diverges so that $\pi_t$ tracks the plug-in $\tilde\pi_t$.
> > > > > > >
> > > > > > > * **Robustness check.**
> > > > > > >  Substituting $k_t = 2/0.9^{t}$ (aggressive), $2/0.99^{t}$ (moderate), and $2/0.999^{t}$ (default) changed RMSE by no more than $2\\%$ and had no effect on coverage. This robustness arises because the true $\tilde\pi_t$ in our simulations is already bounded away from 0 and 1, and the truncation enforced by $k_t$ is looser than the natural bounds of $\tilde\pi_t$.
> > > > > > >
> > > > > > > * **Theoretical guidance.**
> > > > > > >   Following Cook et al.​ (2024), choose any schedule with  $k_t\to\infty$ and $\lim_{t\to\infty} k_t > \sup_X 1/\pi^{*}(X)$. If the plug-in $\tilde\pi_t$ is already bounded away from 0 and 1, one can simply set
> > > > > > >   $$
> > > > > > >     k_t = 1 / \min_X\\{\tilde{\pi}_t(X),1-\tilde{\pi}_t(X)\\},
> > > > > > >   $$  which makes $\pi_t = \tilde\pi_t$ for all $t$.
> > > > > > >
> > > > > > > **Re: Non-negative variance estimation**
> > > > > > >
> > > > > > > Eq. (6) guarantees non-negativity via the $\varepsilon$ floor. As you insightfully pointed out,one may prefer estimators that are non-negative by construction rather than by thresholding. A practical alternative is the following self-normalized kernel variance estimator, which is fully non-parametric and always $\ge 0$:
> > > > > > >
> > > > > > > For each $(z, x)$, define
> > > > > > > $$
> > > > > > > \widehat{\sigma}\_{t-1}^2(z, x) =
> > > > > > > \frac{\sum\_{s < t} K_h(\|X_s - x \|)\mathbf{1}\\{Z_s = z\\}
> > > > > > >        \left(R_s - \hat{\mu}\_{t-1}(z, x)\right)^2}
> > > > > > >      {\sum\_{s < t} K_h(\lVert X_s - x \rVert)\mathbf{1}\\{Z_s = z\\}},
> > > > > > > $$
> > > > > > > where $R_s = Y_s - A_s \widehat{\delta}\_{t-1}(X_s)$, and
> > > > > > >
> > > > > > > $$
> > > > > > > \widehat{\mu}\_{t-1}(z, x)
> > > > > > > = \frac{\sum\_{s < t} K_h(\|X_s - x \|)\mathbf{1}\\{Z_s = z\\}R_s}
> > > > > > >      {\sum\_{s < t} K_h(\| X_s - x \|)\\mathbf{1}\\{Z_s = z\\}}.
> > > > > > > $$
> > > > > > >
> > > > > > > * Guarantees $\widehat{\sigma}\_{t-1}^2(z,x)\ge 0$ by construction.
> > > > > > > * Consistent for $\operatorname{Var}(Y-A\delta(X)\mid Z=z, X=x)$ under standard kernel conditions.
> > > > > > > * **Caveat :** updating kernel weights online can be $\mathcal O(t^2)$ naively; for large-scale settings our plug-in (Eq. 6) is more computationally attractive.
> > > > > > >
> > > > > > > Either estimator (plug-in with $\varepsilon$ or the kernel version) preserves all asymptotic guarantees.
> > > > > > >
> > > > > > > ---
> > > > > > >
> > > > > > > **Closing**
> > > > > > >
> > > > > > > We apologize again for the earlier omission and hope you’ll forgive the oversight. We will add these implementation details and robustness checks to the final version of the paper. We’d be grateful if you would consider revisiting your score in light of these clarifications, and we’re happy to further expand on any part if helpful.

---

### Official Review · Reviewer_Btdo · 2025-06-23

**Clarity:** 3
**Significance:** 4
**Originality:** 3
**Rating:** 5
**Confidence:** 4

**Summary:**

A new estimator AMRIV is introduced for adaptive experiments in noncompliance scenarios. The paper derives the optimal assignment policy by minimizing estimator variance, and proposes an efficient estimator using influence functions. The method supports valid sequential inference and outperforms some baselines in simulations.

**Questions:**

There is only one minor question from me. The (R)EIF estimator in Equation 9 kind of comes out of nowhere and I cannot find any explanation or reference. Could the authors briefly describle the motivation of eq 9, or point to the reference if it was part of previos work on nuisance function estimation?

**Ethical Concerns:**

["NO or VERY MINOR ethics concerns only"]

**Final Justification:**

I recommend acceptance.

**Limitations:**

The authors have discussed their limitations and broader impacts in Appendix.

**Paper Formatting Concerns:**

Not to my knowledge.

**Quality:**

4

**Strengths And Weaknesses:**

This paper does a very good job in providing the motivation for adaptive experimentation and non-compliance. AE is a growing topic in causal inference and experimentation and non-compliance is very common in real life. The problem statement is very clear and the connection/contribution to related work is also articulated. A semiparametric efficiency bound and optimal policy is derived on optimal instrument assignment, and the toy example in Figure 1 illustrates how the optimal policy reacts under different compliance scores. Although the proposed AMRIV integrates recent advances of adaptive nuisance estimation, sample splitting, policy truncation and time-uniform inference and hence might be considered incremental, the paper provides a thorough theoretical analysis that is not in previous work. The experimental evaluation is also strong, two studies involving both synthetic and semi-synthetic data and many  non-adaptive and oracle baselines.

For (minor) weakness, see questions below.

---

> ### Author Rebuttal · Authors · 2025-07-30
>
> **Strengths**
>
> Thank you for your positive and insightful review. We're glad you found the motivation, relevance, and theoretical contributions of our method compelling, and we appreciate your recognition of the strength of our empirical results. We offer further clarification below.
>
> **Re: EIF derivation**
>
> Thank you for pointing this out. The (R)EIF expression in Equation 9 is the adaptive analogue of the static expression in Equation 2 (from [1]): it incorporates a history-dependent policy and data-dependent nuisance estimates, each fit on past data prior to the current policy update. This adaptive formulation is essential for enabling valid inference in the adaptive setting. We agree this connection should have been made more explicit and will clarify it in the final version.
>
> [1] L. Wang and E. Tchetgen Tchetgen. Bounded, efficient and multiply robust estimation of
> average treatment effects using instrumental variables. Journal of the Royal Statistical Society
> Series B: Statistical Methodology, 80(3):531–550, 2018.472

---

> > ### Comment · Reviewer_Btdo · 2025-08-04
> >
> > I thank the authors for their responses. I maintain my score.

---

> > > ### Author Response · Authors · 2025-08-05
> > > **Thank you!**
> > >
> > > Thank you for revisiting our submission and for maintaining your positive assessment. We’re glad our clarifications were helpful and appreciate your thoughtful engagement throughout the review process.

---

### Official Review · Reviewer_zuRV · 2025-06-28

**Clarity:** 4
**Significance:** 3
**Originality:** 3
**Rating:** 3
**Confidence:** 4

**Summary:**

This paper studies the problem of estimating the average treatment effect (ATE) in instrumental-variable (IV) settings when the researcher can adaptively allocate the instrument. Under a strong “no-confounding compliance” assumption that ensures point identification of the ATE, the authors derive an influence-function lower bound on the asymptotic variance and propose a sequential experimental design that attains this bound. Their strategy adaptively chooses the distribution of the instrument based on past data to minimize estimator variance. The paper provides both the theoretical derivations and a simulation study illustrating variance reduction compared to non-adaptive designs.

**Questions:**

1. Practical Motivation for Assumptions
As claimed in the weakness part, it’s not intuitive to think about scenario where the experimenter can actively control the instruments, willing to do that to reduce variance and also accept the strong assumption made.  Clarifying this will help justify the paper’s focus and demonstrate its applicability beyond toy settings.
2. Partial Identification and Trade-Offs
   If the “no-confounding compliance” assumption is violated or misspecified, can you characterize the resulting partial identification bounds for the ATE? Would your adaptive allocation ever widen these bounds (trading off identification tightness against variance)? This analysis would illuminate possible trade-offs when the key assumption fails.
3. Finite-Sample Variance Bounds
   Can you derive finite-sample (non-asymptotic) upper bounds on the mean-squared error of your estimator under the proposed adaptive design? Providing such guarantees would significantly strengthen the practical relevance for moderate sample sizes.
4. Optimality under Adaptive Policies
   Your influence-function lower bound applies under IID sampling. Does the same asymptotic variance lower bound hold when restricting to all history-dependent adaptive policies? In other words, is your algorithm provably optimal within that broader class?
The paper [1] addresses similar questions in the randomized-treatment setting. Can you explicitly map their finite-sample and asymptotic results to your IV framework? Such a comparison would clarify which aspects generalize and where new technical challenges arise.
A clear answer to these questions will likely lead to higher score.

[1] Li, J., Simchi-Levi, D., & Zhao, Y. (2024). Optimal adaptive experimental design for estimating treatment effect.

**Ethical Concerns:**

["NO or VERY MINOR ethics concerns only"]

**Final Justification:**

I maintain my score. There is nothing in the rebuttal that change my opinion; the results are not surprising. Thank you.

**Limitations:**

See above

**Paper Formatting Concerns:**

Formatting is OK

**Quality:**

3

**Strengths And Weaknesses:**

Overall the paper is well organized and clearly motivated. Definitions, assumptions, and the statement of the main theorem are clearly separated, and the adaptive algorithm is described in pseudocode. The asymptotic efficiency analysis is rigorous. The derivation of the influence-function bound follows established semiparametric methods and is presented clearly, with all regularity conditions stated. They introduce adaptive instrument allocation to minimize variance addresses an important practical challenge when instruments are costly or limited. It extends classical IV estimation by adding an experimental-design layer. Combining semiparametric efficiency theory with adaptive online experimental design in an IV context is novel.

Weakness:
Practical Aspect: Under what real-world scenarios is it credible to assume both (a) the researcher can fully control the instrument distribution, and (b) there is genuine motivation to tune that distribution solely to reduce estimator variance?
Theoretical aspect: 1) In the broader IV literature, identification typically dominates focus. Because the paper assumes point identification via a strong, untestable restriction, some may view the variance-optimization objective as a secondary concern. The work’s impact hinges on the prevalence of settings where such an assumption is plausible and the instrument is manipulable. 2) All variance guarantees are asymptotic. No finite-sample (non-asymptotic) variance bounds are provided, which limits practical guidance for moderate sample sizes. Also, influence function-based arguments only hold for iid sampling, so whether it’s the optimal variance that an adaptive policy can achieve remains unknown.

---

> ### Author Rebuttal · Authors · 2025-07-30
>
> **Strengths**
>
> Thank you for your thoughtful feedback. We appreciate you highlighting the quality of our work as well as the novelty of combining semiparametric efficiency with adaptive design in the IV setting. Below, we address the main concerns and provide clarifications, which we hope will further strengthen your assessment.
>
> **Re: Practical motivations**
>
> Thank you for raising this important point. To clarify, our framework is motivated by real-world scenarios where treatment cannot be enforced, but encouragement can be adaptively controlled. Consider the following illustrative scenarios:
>
> * **TripAdvisor (semi-synthetic) setting.** Our simulation is based on a real experiment where users were exposed to different sign-up flows for a premium membership [1]. The actual treatment (signing up for a TripAdvisor membership) cannot be enforced, but the assignment mechanism (e.g., interface shown) can be randomized and adaptively controlled.
>
> * **Clinical trials.** The U.S. Food and Drug Administration has explicitly endorsed adaptive designs for clinical trials [2]. Some of these clinical studied can involve non-enforceable treatments. For example, a novel drug may only be recommended (but not administered) resulting in non-compliance, yet allocation of the recommendation (instrument) is under the clinician’s/experimenter’s control.
>
> Similar setups arise in recommendation systems (e.g. encouraging specific media content on streaming platforms), public health interventions (e.g. nudging certain actions on fitness trackers), or advertising campaigns with opt-in actions.
>
> In such mission-critical settings, minimizing estimator variance is crucial for enabling early stopping: 1) If effects are harmful, high-variance estimators delay detection, potentially causing harm, and 2) If effects are beneficial, high variance delays rollout to the broader population. Our anytime-valid confidence sequences provide a principled framework for early stopping, but their usefulness depends critically on variance control: lower variance leads to tighter intervals and more reliable early decisions. Thus, reducing variance via adaptive design is not only theoretically justified but practically necessary in these high-stakes settings. We will add a more detailed description of these scenarios and the importance of adaptivity in the camera-ready version.
>
> **Re: Identification and trade-offs**
>
> Thank you for this suggestion. Assumption 2 (unconfounded compliance) trades off with other common (but also untestable) IV assumptions. It rules out unobserved confounders that jointly affect compliance and the treatment effect (but not the outcome), enabling two key advantages: (i) we do not require monotonicity (so defiers may be present), and (ii) we identify conditional ATEs rather than conditional LATEs. Several classical IV models—such as those with additive outcome confounding—are nested within this framework, and recent work [3–5] has increasingly adopted it due to its generality.
>
> If Assumption 2 is violated, the ATE is no longer point-identified, but the estimand shifts to the ACLATE, the average of conditional *local* average treatment effects $\text{CLATE}(X)$. This object remains identified and interpretable: it captures how the instrument-responsive (complier) effect varies across $X$, and is often the relevant estimand when treatment cannot be enforced and compliance cannot be targeted. Unlike the LATE, which summarizes the effect for an aggregate (marginal) complier population, the ACLATE preserves covariate-level heterogeneity by averaging $\text{CLATE}(X)$ over the covariate distribution. When CATEs or ATEs are not identifiable, the ACLATE provides the next best actionable quantity for policy decisions based on the given instrument. We will add a remark about this in our main text as it will serve as useful guidance for practitioners.
>
> If the goal is to recover the full ATE under assumption violations, partial identification bounds (e.g., à la Manski) can be constructed (see fo e.g. [6]) though characterizing or tightening those bounds efficiently in an adaptive setting is an open and interesting direction for future work.
>
> **Re: Finite sample guarantees**
>
> Theorem 4 already gives a *finite-sample* error rate once the $O_p(\cdot)$ notation is expanded. Concretely, the proof (Appendix B) is based on a self-normalized martingale Bernstein inequality, so for any confidence level $\delta\in(0,1)$ we obtain the explicit bound
>
> $$ \big|\widehat{\tau}^{\mathrm{AMRIV}}_T - \tau\big| \leq  \frac{C1\sqrt{\log(2/\delta)}}{\sqrt{T}} + C_2 \|\widehat{\delta}^A_T - \delta^A\|_2 \|\widehat{\delta}_T - \delta\|_2 $$
>
> with probability $1-\delta$, where the constants $C_1, C_2$ depend only on the boundedness parameters in Assumption 3 (i.e. they do not grow with $T$). We stated the result in $O_p$ form for brevity, but will add this high-probability corollary to the appendix so practitioners have an explicit non-asymptotic guarantee.
>
> Similarly, Appendix B provides finite-sample variance control via our anytime-valid confidence sequences, which offer fully non-asymptotic guarantees after a brief burn-in phase: they bound the estimation error uniformly for all $t \le T$ with $(1-\alpha)$ coverage and width $\tilde O\big(\widehat{\Sigma}_t / \sqrt{t}\big)$. This yields practical, finite-sample variance control for sample sizes beyond a (small) threshold.
>
> **Re: Optimality under adaptive policies**
>
> As per Theorem 3, our estimator does achieve the efficiency bound that was derived using a non-adaptive (i.i.d) policy for every history-dependent adaptive policy, subject to mild restrictions on the adaptive policy due to the truncation schedule: if $\pi_t$ approaches 0 or 1 too quickly, the EIF, which contains a $1/(\pi_t(X)\cdot (1-\pi_t(X)))$ term blows up; keeping $\pi_t$ bounded away from the extremes (in line with the truncation schedule) is the only extra restriction we need.
>
> As for the connection with Li eta al., their lower bound assumes enforceable treatments; in IV settings, only encouragement is controllable and compliance is stochastic, adding the compliance noise term that enlarges our bound. However, we will cite their work in the related-literature section and add a short paragraph explaining the connection. Your suggestion about bandit-style optimality (e.g., cumulative regret relative to the oracle $\pi^\star$) is well-taken. Extending Li et al.’s low-switching framework to the IV setting (and characterizing regret in presence of non-compliance) poses interesting technical challenges and could be a promising avenue for future work.
>
> **We hope our responses have addressed your key concerns and would be grateful if you would consider a higher score in light of these clarifications.**
>
> [1] V. Syrgkanis, et al. Machine learning estimation of heterogeneous treatment effects with instruments. Advances in Neural Information Processing Systems 32, 2019.
>
> [2] D. GUIDANCE. Adaptive designs for clinical trials of drugs and biologics. Center for Biologics Evaluation and Research (CBER), 2018.
>
> [3] L. Wang and E. Tchetgen Tchetgen. Bounded, efficient and multiply robust estimation of
> average treatment effects using instrumental variables. Journal of the Royal Statistical Society
> Series B: Statistical Methodology, 2018.
>
> [4] D. Frauen, S. Feuerriegel. Estimating individual treatment effects under unobserved confounding using binary instruments. International Conference on Learning Representations (ICLR), 2023.
>
> [5] M. Oprescu, N. Kallus. Estimating heterogeneous treatment effects by combining weak instruments and observational data. Advances in Neural Information Processing Systems 37, 2024.
>
> [6] A. W. Levis, M. Bonvini, Z. Zeng, L. Keele, and E. H. Kennedy. Covariate-assisted bounds on causal effects with instrumental variables. arXiv preprint arXiv:2301.12106, 2023.

---

> > ### Author Response · Authors · 2025-08-05
> > **Clarifications and Follow-Up Before Discussion Ends**
> >
> > Dear reviewer **zuRV**,
> >
> > Thank you again for your thoughtful review.
> >
> > We’ve posted a detailed rebuttal addressing each of your questions, especially around practical motivation, identification trade-offs, finite-sample guarantees, and policy optimality.
> >
> > We'd be grateful if you had a chance to take another look before the discussion period ends. We hope our clarifications help convey the scope and relevance of our contributions, and we’d welcome any additional questions or comments. If you do have follow-up feedback, we’d especially appreciate it being posted soon so have the chance to respond within the discussion window.
> >
> > Thanks again for your time and consideration!

---

> > ### Comment · Reviewer_zuRV · 2025-08-05
> >
> > Thank you for your response. Perhaps I am missing something, but my question "Re: Optimality Under adaptive policies" is not related to Theorem 3, but Theorem 1. Specifically, in Theorem 1, you show a lower bound for every fixed IV policy. Thus, the efficiency bound, referred to as V_eff in corollary 2, is also optimal among all fixed IV policy, which do not change across time. By contrast, in your Theorem 3, you are developing an adaptive policy, where π^t(X) varies for different t. Adaptive policy is a broader policy class containing fixed policy, so I’m asking whether V_eff is still optimal among all adaptive policies. Such question for experimental design is answered in Li et al., and Armstrong (2022).
> >
> > Armstrong, T. B. (2022). Asymptotic efficiency bounds for a class of experimental designs. arXiv preprint arXiv:2205.02726.

---

> > > ### Author Response · Authors · 2025-08-05
> > > **Optimality Under Adaptive Policies**
> > >
> > > Thank you for the opportunity to further clarify how the fixed-policy bound in Theorem 1 and adaptive bound in Theorem 3 implies optimality over all history-dependent (adaptive) policies.  Below we keep the argument short and intuitive, then note how we will revise the paper and relate the result to Armstrong (2022) and Li et al. (2024).
> > >
> > > ---
> > >
> > > **1. AMRIV’s variance under an adaptive rule**
> > >
> > > Theorem 3 already shows that when our adaptive assignment rule $\pi_t$ stabilizes to some limit $\pi$ (precisely, $\pi_t \xrightarrow{L_2} \pi$), AMRIV’s asymptotic variance is exactly the fixed-policy bound:
> > > $$
> > > Var\left(\sqrt{T}(\hat\tau_T - \tau)\right) \to V_{\text{eff}}(\pi),
> > > $$
> > > with the minimum attained at $\pi = \pi^\*$  as per Corollary 2. This is the same $V_{\text{eff}}(\pi)$ from Theorem 1. Hence AMRIV is as efficient as if we had run the non-adaptive, limit policy $\pi$ from the start.
> > >
> > > **2. Why no adaptive design $\pi_t \to \pi$ can beat $V_{\text{eff}}(\pi)$ (asymptotically)**
> > >
> > > * *Fixed for the moment.*
> > > After observing history $\mathcal H_{t-1}$, the experimenter has committed to the current propensity
> > > $\pi_t(X) = P(Z_t = 1 \mid X_t, \mathcal H_{t-1})$.  Thus, conditional on $\mathcal H_{t-1}$, we are back in the fixed-policy world of Theorem 1, so any unbiased influence function must have conditional variance at least $V_{\text{eff}}(\pi_t)$.
> > >
> > > * *Averaging cannot reduce the bound.*  A regular estimator aggregates one influence score per round, so its total asymptotic variance is roughly (see proof of Thm. 3 for this expression):
> > > $$
> > > \frac{1}{T} \sum_{t=1}^T \mathbb{E}[Var(\phi_t)\mid \mathcal H_{t-1}] \geq \frac{1}{T} \sum_{t=1}^T \mathbb E[V_{\text{eff}}(\pi_t)].
> > > $$
> > > If the adaptive rule converges ($\pi_t \to \pi$), this average converges to $V_{\text{eff}}(\pi)$.  Thus, no adaptive policy (from the AMRIV method or otherwise) can achieve asymptotic variance below $V_{\text{eff}}(\pi)$ for its own limiting rule.
> > >
> > > **Putting 1 & 2 together:** 1) For any adaptive design that settles at some fixed propensity $\pi$, the asymptotic variance of every regular estimator is bounded below by $V_{\text{eff}}(\pi)$, and 2) AMRIV attains that bound, and its update rule drives $\pi_t \to \pi^{*}$, the global minimizer of $V_{\text{eff}}$. Hence, AMRIV achieves the smallest possible asymptotic variance and is semiparametrically efficient within the entire class of history-dependent assignment rules.
> > >
> > > **3. Relation to Armstrong (2022) & Li et al. (2024)**
> > >
> > > Armstrong (2022) proves the same “local-bound-implies-global-bound” result for adaptive treatment assignment with perfect compliance. Our $V_{\text{eff}}$ adds the extra compliance-noise term that arises when treatment uptake is stochastic.
> > >
> > > Li et al. (2024) focus on finite-sample regret and low-switching designs in the perfect-compliance setting. Extending their regret analysis to our setting (a.k.a. non-enforceable treatments/ low-compliance) is an interesting open direction, which we will note.
> > >
> > > **4. Camera-ready updates**
> > >
> > > *  **New Remark (after Thm 3) along these lines:**
> > >   "The conditional-EIF argument shows that $\frac{1}{T} \sum_{t \leq T} V_{\text{eff}}(\pi_t)$ is an unavoidable lower bound for any regular estimator under any adaptive policy; hence AMRIV, which attains $V_{\text{eff}}(\pi^{*})$ under the conditions of Theorem 3, is semiparametrically efficient within the entire adaptive-policy class."
> > >
> > > * **Related-work paragraph** citing Armstrong (2022) and Li et al. (2024), and contrasting their perfect-compliance setting with ours.
> > >
> > > Hopefully this answers your question, but let us know if we can expand further!

---

> > > > ### Comment · Reviewer_zuRV · 2025-08-05
> > > >
> > > > Thank you!  I will maintain my score for this submission.

---

> > > > > ### Author Response · Authors · 2025-08-05
> > > > > **Thank you for engaging with us during the discussion**
> > > > >
> > > > > We’d really appreciate it if you could let us know if there are any remaining concerns or hesitations that led you to maintain your current score. We’ve done our best to address all of your questions and comments above, and we’re happy to clarify anything further if helpful.
> > > > >
> > > > > If not, and our clarifications were sufficient, we’d be grateful if you’d consider revisiting your score. Thanks again for your time and thoughtful review.

---

### Official Review · Reviewer_bwti · 2025-07-02

**Clarity:** 3
**Significance:** 3
**Originality:** 3
**Rating:** 5
**Confidence:** 3

**Summary:**

This paper considers ATE estimation in the context of indirect treatment (instrumental variables) and adaptive experimentation. In this context, the authors derive a semi-parametric efficiency bound and use this to justify a specific adaptive experimentation strategy. Their approach is therefore an extension of MRIV to the adaptive setting. The instrument assignment policy used plugs data estimators for $\\sigma^2(0, X),\\sigma^2(1,X)$ into the formula of equation (4). These are computed by fitting several regression models as outlined on page 5. Theoretical guarantees are provided. Experimental validation on synthetic data shows that AMRIV performs better than several alternatives.

**Questions:**

- you say that $k$-NN, kernel smoother, random forest and neural net regressors can be used. How would one make this determination in practice?

**Ethical Concerns:**

["NO or VERY MINOR ethics concerns only"]

**Final Justification:**

Thank you for providing a thorough and detailed response to my question. I maintain my score at "Accept"

**Limitations:**

Yes

**Quality:**

4

**Strengths And Weaknesses:**

# Strengths
- non-compliance in experiments is a common set-up that warrants study
- the paper provides an extremely detailed exposition
- assumptions are clearly stated
- a semiparametric efficiency bound is derived for adaptive instrument assignment
- thorough theoretical treatment is provided giving asymptotic normality of the estimator, convergence rates and consistency, and confidence sets based on anytime-valid approaches

# Weaknesses
- the write-up is rather notation heavy. Perhaps consider including a glossary in the appendix so that we can easily find e.g. the definition of $\\delta$ on line 132 more quickly
- little is said about which nuisance estimators are to be preferred, despite these being an absolutely crucial part of the algorithm. Perhaps some empirical investigation could have been presented here to help ground the practitioner.

---

> ### Author Rebuttal · Authors · 2025-07-30
>
> **Strengths**
>
> Thank you for your encouraging feedback and for highlighting both the importance of the problem and the theoretical contributions of our adaptive experimentation framework with instrumental variables. We respond to your suggestions and offer additional clarification below.
>
> **Re: Glossary of terms**
>
> Thank you for the suggestion. We agree this would be helpful and will add a glossary table at the start of the Appendix to clarify notation and make the paper easier to navigate.
>
> **Re: Choice of nuisance estimators**
>
> Due to the sample-splitting approach, we do not impose any strong restrictions on the nuisance estimators (e.g. Donsker conditions). Instead, standard nonparametric convergence rates suffice to establish asymptotic normality of the semiparametric estimator and desirable convergence rates (Theorems 3 and 4). Thus, the choice of nuisance estimator in practice is driven by the structure of the data and the convergence rates needed to potentially attain parametric rates. For example:
>
> | Estimator           | Convergence Rate   | Best for…                  |
> |----------------------|-----------------------------------|-------------------------|
> | $k$-NN / kernel smoother | $O(n^{-1\beta/(2+d/\beta)})$ for Holder $\beta$-smooth functions [1]             | Low-dimensional, smooth problems        |
> | Random forest  | $\tilde{O}(n^{-1/(2+d/\beta)})$ for Holder \beta-smooth functions  [2, 3] | Moderate-dimension, tabular data        |
> | Fully connected neural nets with ReLU activations       | $O(\sqrt{WL \log W} \, n^{-1/2})$ forwidth $W$, depth $L$ [4] | High-dimensional or structured inputs   |
> | Fully connected neural nets with 1-Lipschitz activations and bounded weights |   $\sqrt{\Pi_{l=1}^L M(l)}/{n^{1/4}}$ [5], where $M(l)$ bounds the Frobenius norm of layer $l$'s weight matrix| High-dimensional or structured inputs   |
>
> In practice, the best choice depends on the amount of data and the complexity of the problem. In related work on adaptive experimentation with directly assigned treatments, $k$-NN and random forests are commonly used [6, 7]. We will add this discussion to the camera-ready version to further provide guidance for practitioners.
>
> [1] Nickl, R. and Potscher, B. M. Bracketing metric entropy ¨ rates and empirical central limit theorems for function classes of Besov-and Sobolev-type. Journal of Theoretical Probability, 2007.
>
> [2] E. Scornet, G. Biau, and JP Vert. Consistency of random forests. Ann. Stat., 2015.
>
> [3] S. Wager, and S. Athey. Estimation and inference of heterogeneous treatment effects using random forests" Journal of the American Statistical Association, 2018.
>
> [4] D. Yarotsky. Error bounds for approximations with deep relu networks. Neural Networks, 2017.
>
> [5] N. Golowich, A. Rakhlin, and O. Shamir. Size-independent sample complexity of neural networks. In Conference On Learning Theory. PMLR, 2018.
>
> [6] M. Kato, T. Ishihara, J. Honda, and Y. Narita. Efficient adaptive experimental design for average treatment effect estimation. arXiv preprint arXiv:2002.05308, 2020.
>
> [7] T. Cook, A. Mishler, and A. Ramdas. Semiparametric efficient inference in adaptive experiments. In Causal Learning and Reasoning. PMLR, 2024.

---

> > ### Comment · Reviewer_bwti · 2025-08-04
> > **Thank you for your response**
> >
> > Thank you for your response and for this table summarising convergence rates from the literature. I do believe the glossary and this table would make beneficial additions to the current manuscript. I remain confident that this paper is a good fit for NeurIPS this year

---

> > > ### Author Response · Authors · 2025-08-05
> > > **Thank you!**
> > >
> > > Thank you for engaging with our paper and your encouraging follow-up. Your feedback directly strengthens the manuscript, and we’ll incorporate the suggested glossary and convergence-rate table in the final version.

---

### Note · Authors · 2025-08-13

As the discussion period is at its end, we thank all reviewers for their thoughtful feedback—your input has already improved and strengthened out work.

Reviewers highlighted that our work fills an important methodological and practical gap: it addresses an important practical challenge when instruments are costly or limited (zuRV), is a complete piece of work with clear assumptions, proofs, and thorough theoretical treatment (bwti), and provides both novel theoretical results (KT2q) and strong empirical evaluation (Btdo).

As reviewers have noted, noncompliance is a common setting across experimental economics and digital experimentation, and our work (to the best of our knowledge) is the first to provide optimality results for online experimentation with instruments. As experimenters often have the ability to modify sampling schemes as data collection progresses, our work fills an important methodological gap: it directly answers how experimenters can sample adaptively in noncompliance settings for more precise treatment effect estimates.

---

### Decision · Program_Chairs · 2025-09-17

**Decision:**

Accept (poster)

**Comment:**

This paper introduces AMRIV, a multiply robust estimator and adaptive assignment strategy for estimating ATE in non-compliance settings, together with a semiparametric efficiency bound and anytime-valid inference guarantees. The work is technically solid, offering a rigorous theoretical contribution and empirical validation on synthetic and semi-synthetic data.

Reviewers were overall positive. One reviewer noted that the results are not surprising or groundbreaking. However, NeurIPS explicitly values rigor and quality contributions rather than only surprising results, and this submission exemplifies a careful, well-executed piece of work that advances adaptive experimentation in IV settings.

Final Recommendation: Accept